# Evaluating place cell detection methods in Rats and Humans: Implications for cross-species spatial coding

Weijia Zhang[1]*, Thomas Donoghue[1], Salman E. Qasim[2], Joshua Jacobs[3]*

1 Department of Biomedical Engineering, Columbia University, New York, New York, United States of America, 2 Department of Neurosurgery, Rutgers Robert Wood Johnson Medical School and Rutgers Brain Health Institute, New Brunswick, New Jersey, United States of America, 3 Department of Neurology and Neuroscience Institute, University of Chicago, Chicago, Illinois, United States of America

* wz2540@columbia.edu (WZ); joshuajacobs@uchicago.edu (JJ)

## Abstract

Place cells, first identified in the rat hippocampus as neurons that fire selectively at specific locations, are central to investigations of the neural underpinnings of spatial navigation. Recent spatial studies in human patients with drug-resistant epilepsy have made identifying and characterizing place cells across species increasingly important for understanding the extent to which decades of rodent research generalize to humans and for uncovering fundamental principles of spatial cognition. One challenge, however, is that detection methods differ: rodent studies often rely on spatial information (SI) in conjunction with place field stability measures, whereas human studies employ analysis of variance (ANOVA) based approaches. These methodological differences may affect the identified place cell populations, which complicates how their properties are interpreted and cross-species comparisons. To address this, we systematically applied multiple detection pipelines to human and rat datasets, supported by simulations that vary place-field properties. Our analyses and simulations demonstrate that spatial information and ANOVA-based approaches are responsive to distinct place field properties: spatial information primarily reflects the contrast between peak and average firing rates, while ANOVA emphasizes consistency across trials. Across species, rodent place cells revealed a broad spectrum of spatial tuning, including strongly tuned neurons with high spatial information and high ANOVA values. In contrast, human place cells lacked this strongly tuned population and exhibited a narrower distribution of tuning scores, concentrated at the lower end of both spatial tuning metrics. Despite these differences, both species had an overlapping population of neurons with weaker yet consistent spatial tuning, which may support important functional roles such as generalization and mixed selectivity. Addressing these analytical differences allows for more direct comparisons between species, though differences in spatial tuning may still relate to variations in experimental paradigms that warrant further investigation. Together, our study provides

**Data availability statement:** Repository This project is openly available through an online project repository, which includes all the code used for data pre-processing and analysis. Project Repository: https://github.com/HSNPipeline/PlaceCellMethods Dataset This project uses electrophysiological data collected from neurosurgical patients, as well as an open access rat recording dataset from CRCNS. org: http://dx.doi.org/10.6080/K09G5JRZ. The human single-neuron dataset was collected as part of a previously published study and is publicly available through OSF [14]: https://osf.io/dh3wv/metadata/osf. To systematically evaluate place cell detection methods across species, we developed a custom simulation framework, SimPlaceCells, available at: https://github.com/HSNPipeline/SimPlaceCells. Software All code used and developed for this project was written in the Python programming language. The code is openly available, licensed for reuse, and deposited in the project repository. Management of the dataset was conducted using the Human Single Neuron (HSN) Pipeline: https://github.com/HSNPipeline Analyses of the single-neuron data were performed using the open-source Spiketools toolbox: https://github.com/spiketools/spiketools Literature searches and related resources were organized using LISC, an open-source Python module for literature analysis. https://github.com/lisc-tools/lisc.

**Funding:** This work was supported by NIH grant R01MH104606 to JJ, which provided salary support for WZ & TD. The funders had no role in study design, data collection and analysis, decision to publish, or preparation of the manuscript.

**Competing interests:** The authors have declared that no competing interests exist.

a roadmap showing how spatial tuning metrics shape place cell detection and interpretation.

---

## Author summary

Place cells are neurons that become active in specific locations, and they play a critical role in how the brain supports navigation and memory. Place cells were first discovered in rats and later observed in humans, however, there has been a lack of direct comparisons between species using comparable approaches. Part of the difficulty of doing so is that studies of rodent and human place cells have often relied on different analysis methods, making it difficult to determine if and how place-cell properties differ between species. To address this, in this study, we set out to understand how differences in place cell detection methods affect the identified place cell populations and interpretations of spatial coding across species. To do so, we compared the most prevalent detection methods used in rodent and human research side by side, applying them to datasets from both species and to simulations. We found that different methods emphasize different features of spatial responses, which changes which neurons are identified as place cells. Across species, rat recordings revealed a wide range of spatial responses, from neurons with sharply localized activity to those with broader but reliable patterns. Human recordings, by contrast, were more concentrated at weaker but consistent levels of tuning. Importantly, these weaker but consistent responses reflect an overlapping population of neurons found in both species, which may serve similar functional roles in supporting flexible spatial memory and generalization. By separating methodological effects from biological differences, we lay the groundwork for future cross-species studies for spatial coding.

## Introduction

The hippocampus plays a central role in spatial navigation and memory by forming internal representations of the environment's structure and the animal's experiences. A hallmark of this function is the activity of place cells, neurons that fire selectively when an animal occupies a specific location [1,2]. First discovered in rodents, place cells exhibit tuning properties that vary systematically with anatomical location and behavioral demands. In dorsal CA1, rat place cells exhibit sharply localized and stable firing fields that tile the environment with high spatial precision [3–6]. Moving along the hippocampal axis towards the ventral regions, neurons tend to exhibit broader and more diffuse fields and greater responsiveness to motivational and contextual variables [6–9].

In humans, analogous spatially modulated neurons have been identified through intracranial recordings from neurosurgical patients with drug-resistant epilepsy performing virtual navigation tasks. These neurons increase firing at specific locations,

supporting a conserved role in spatial representation [10–13]. Human neurons exhibit spatial tuning with relatively diffuse place-field boundaries, and their firing patterns are often also modulated by non-spatial factors such as task demands, contextual cues, and goal relevance [14–20]. While place cells have been described in both humans and rodents, the human literature is comparatively smaller, and few studies have directly examined their properties across species using comparable behavioral paradigms. As a result, it remains unclear to what extent place-cell properties differ between species and what factors give rise to these differences.

Methodological approaches used to identify place cells may play a critical role in shaping our understanding of their properties. In this study, we evaluate place cells as neurons that exhibit spatial tuning as determined by our statistical criteria, and we clarify how different analytical methods relate to recommended best practices. Establishing this criterion provides a consistent foundation for comparing analytical approaches and interpreting results across species. Previous methodological work has shown that the choice of detection metric influences which neurons are considered spatially tuned, but these comparisons have largely been confined to a single species or a single analytic framework [21,22]. Rodent studies often identify spatial tuning using spatial information (SI) scores, which quantify how much neural firing reduces uncertainty about position [23]. SI thresholds (e.g., 0.25–0.5 bits/spike) or permutation-based significance tests are typically used in conjunction with stability measures, such as odd–even correlations, as well as additional criteria like minimum firing rate, to classify place cells (Table 1)

Applying a threshold (e.g., 0.25–0.5 bits/spike) or a permutation-based significance test is often combined with additional criteria, such as minimum firing rate and trial-level stability, to classify place cells (Table 1) [9, 24–30]. Human studies typically rely on analysis of variance (ANOVA) measures, evaluated against permuted surrogates, to assess whether firing rates vary significantly across spatial bins, often incorporating additional task-related modulations (Table 2) [10,17,19,31]. These methodological differences may influence which neurons are identified as place cells and, in turn, affect how their properties are interpreted, underscoring the need to clarify how methodological choices impact place-cell research across species.

Place cells are typically defined as neurons that exhibit spatially selective firing when the animal occupies a specific location [1]. Here, we take a broad perspective on place cell activity, employing statistical detection methods that capture spatial tuning, including neurons with classical, sharply tuned, place fields, as well as more diverse spatial response patterns (e.g., responses that may have broader or more variable responses, as long as they exhibit in a way that is captured by the detection methods). By using this approach, we hope to clarify how different analytical methods reflect the underlying properties of place cells, informing best practices for place-cell detection across species. To this end, we perform a systematic evaluation of place cell detection methods using both empirical datasets (rat and human single-neuron recordings) and simulate datasets with known ground truth. We focus on two widely used spatial tuning metrics: spatial information and ANOVA F-statistics. We assess how both raw tuning scores and downstream classification criteria, including fixed thresholds and permutation-based significance tests, influence the identification of place cells. To further investigate how each spatial tuning metric relates to the underlying properties of place fields, we develop a simulation framework that systematically varies key place field properties such as tuning width, baseline firing rate, and trial-to-trial variability. Finally, we extract estimated place-cell features from rat, human, and simulated data and projected them into a low-dimensional representational space, revealing how spatial tuning metrics differentially reflect place-field features and their interactions.

Our results suggest that commonly used spatial tuning metrics: spatial information (SI) and ANOVA capture different features of place cell activity. SI is more responsive to sharply localized, high-contrast firing fields, whereas ANOVA emphasizes the trial-level consistency of spatial tuning curves. Importantly, identified place cell populations depend strongly on the choice of classification criteria: raw score thresholding and permutation-based significance testing often yield divergent results, with the latter offering more stable and data-adaptive criteria. We also find that rat place cells exhibit a broader dynamic range of spatial tuning scores than humans. Rodent neurons frequently display sharply localized place fields with high spatial specificity, resulting in cells that score highly on both SI and ANOVA metrics. In contrast,

**Table 1. Classification criteria used in rodent place-cell studies.**

| Reference | Spatial Metric | Classification Criteria | Modulation |
|---|---|---|---|
| Jung et al., 1994 [7] | SI[1]; Sparsity[2] | Place Field[3] | Place; Anatomy |
| Schapiro et al., 1997 [32] | N/A | Place Field[3] | Place; Cues |
| Wood et al., 2000 [33] | Two-Way ANOVA[4] | Place Field[3]; Parametric ANOVA main effect of spatial sector ($P < 0.05$) | Place; Trial Type |
| Kjelstrup et al., 2008 [34] | SI[1] | Place field[3] | Place; Anatomy |
| Royer et al., 2010 [35] | SI[1]; Stability[5.1] | Place field[3] | Place; Anatomy |
| Langston et al., 2010 [25] | SI[1]; Spatial Coherence[6] | Permutation-based SI[1] score ($P < 0.05$); Spatial Coherence $< 0.14$ | Place; Development |
| Deshmukh et al., 2011 [26] | SI[1]; Stability[5.3] | SI[1] $> 0.4$; Permutation-based SI[1] score ($P < 0.01$); Stability $> 0.71$ | Place; Object |
| Chen et al., 2013 [27] | SI[1] | Permutation-based SI[1] score ($P < 0.05$); PTD[7] $> 300\,\mu s$ | Place; Visual |
| Roux et al., 2017 [24] | SI[1]; Stability[5.3] | SI[1] $> 0.25$; Permutation-based SI[1] score ($P < 0.05$) | Place; Learning |
| Scalpen et al., 2017 [28] | SI[1] | SI[1] $> 0.25$; Place Field[3] | Place; Visual |
| Aronov et al., 2017 [36] | SI[1]; Stability[5.1] | Permutation-based SI[1] score ($P < 0.01$) | Place; Sound |
| Newman et al., 2017 [37] | SI[1] | SI[1] $\geq 0.5$; Place Field[3] | Place; Cholinergic Disruption |
| Jun et al., 2020 [38] | SI[1] | Permutation-based SI[1] score ($P < 0.05$); Place Field[3] | Place; Alzheimer's |
| Grieves et al., 2020 [39] | SI[1]; Stability[5.1] | SI[1] $> 0.5$; PTD[7] $> 250\,\mu s$ | 3D Place |
| Shuman et al., 2020 [40] | SI[1]; Stability[5.1,5.2] | Permutation-based SI[1] score and within-session stability[5.1,5.2] ($P < 0.05$) | Place |
| Duvelle et al., 2021 [29] | SI[1] Stability[5.3] | SI[1] $> 0.5$; Permutation-based SI[1] score ($P < 0.05$); PTD[7] $> 300\,\mu s$ | Place; Connectivity |
| Harland et al., 2021 [30] | SI[1] | SI[1] $> 0.5$ | Place; Environment Size |
| Jin & Lee, 2021 [9] | SI[1]; Stability[5.1] | SI $> 0.5$; Permutation-based SI[1] score ($P < 0.01$) | Place; Motivation |
| Tanni et al., 2022 [41] | Stability[5.1,5.2] | Spike Shape: PTD[8] $> 0.45$ ms; HW[9] $> 0.1$ ms; TR[10] $> 0.175$; Stability[5.1] $> 0.25$, Stability[5.2] $> 0.5$; Place Field[3] | Place; Perceptual Change |
| Levy et al., 2023 [42] | SI[1]; Spatial Coherence[6] | SI[1]; Spatial coherence[6] ($Z > 1.96$ for both) | Place |
| Blair et al., 2023 [43] | Stability[5.1] | Median Stability[5.1] ($P < 0.01$) | Place; Aversive Experience |
| Zhang et al., 2024 [44] | SI[1] Stability[5.1] | Permutation-based SI[1] score ($P < 0.05$); Stability[5.1] $> 0.3$ | Place; Social |

*SI[1]*: Spatial information score (bits/spike). *Sparsity[2]*: Measures how concentrated firing is across space. *Place Field[3]*: A contiguous group of adjacent spatial bins in which the average firing rate in each bin exceeds a specified threshold. *Two-Way ANOVA[4]*: A significant main effect of place was observed, but not for task-related variables or the place × task-related variable interaction. *Stability[5]*: [5.1]Split/Half-Correlation between first and second half maps. [5.2]Even/Odd-Pearson correlation from even/odd lap tuning curves. [5.3]Pixel by pixel correlation coefficient between epochs/sessions. *Spatial Coherence [6]*: Estimated as the first order spatial autocorrelation of the place field map - i.e., the mean correlation between the firing rate of each bin and the averaged firing rate up to 8 adjacent bins. *PTD[7]*: Waveform peak to trough distance. *PTD[8]*: Waveform peak-to-trough duration. *HW[9]*: Waveform peak half-width. *TR[10]*: The ratio between amplitude and trough voltage values.

such sharply tuned and high contrast cells are rare in human data, where hippocampal neurons more commonly exhibit spatially diffuse but trial-consistent tuning, leading to lower overall SI and F-statistic values. Notably, both species show overlapping distributions at the lower end of the SI and ANOVA spectra, suggesting an analogous population of neurons with weaker but reliable spatial modulation - cells that are often missed by SI thresholds but consistently detected by ANOVA. These ANOVA-identified neurons likely contribute to generalized representations and may reflect mixed selectivity for spatial and task-related variables, which are important for understanding spatial coding across species.

Together, these findings indicate that commonly used detection methods capture different but complementary aspects of spatial coding. By clarifying how methodological choices shape the identification of these cells, our work establishes a foundation for principled cross-species comparisons. Broadening our focus to include consistently tuned,

**Table 2. Summary of classification methods and modulation types used to identify place-related neuronal activity in human single-neuron studies.**

| Reference | Spatial Metric | Classification Criteria | Modulation |
|---|---|---|---|
| Ekstrom et al., 2003 [10] | ANOVA[10] | ANOVA F-Statistic above chance: $P<0.05$ | Place + goal + view |
| Jacobs et al., 2010 [11] | One-sided rank-sum test[11] | One-sided rank-sum test with permutation-corrected threshold ($P<0.05$); Place Field[3] | Place + direction; Path |
| Jacobs et al., 2013 [13] | ANOVA[10] / T-test[11] | Location-specific *T*-tests vs. shuffled data ($P<0.05$); direction sensitive place cells are assessed by ANOVA. | Place + direction |
| Miller et al., 2013 [12] | ANOVA[10] / One-sided rank-sum test[11] | ANOVA testing spatial context × recall interaction; one-sided Wilcoxon rank-sum test with permutation-corrected threshold ($P<0.05$); place field[3]. | Place + recall |
| Qasim et al., 2019 [14] | ANOVA[10] | Permutation-based ANOVA F-statistic ($P<0.05$). | Place + object |
| Tsitsiklis et al., 2020 [15] | ANOVA[10] | Permutation-based ANOVA F-statistic ($P<0.05$). | Place; Spatial target |
| Kunz et al., 2021 [16] | ANOVA[10] | Permutation-based ANOVA F-statistic ($P<0.05$) | Place + direction + egocentric |
| Qasim et al., 2021 [17] | ANOVA[10] | Permutation-based ANOVA F-statistic ($P<0.01$). | Place |
| Schonhaut et al., 2023 [18] | OLS regression[13] | Main effects (time, place, navigation event) tested via permutation-based OLS regression ($P<0.05$) | Place + time / events |
| Donoghue et al., 2023 [19] | ANOVA[10] | Permutation-based ANOVA F-statistic ($P<0.05$) | Place; Spatial target |
| Kunz et al., 2024 [20] | T-test[11] | *T*-tests vs. shuffled data ($P<0.05$);Place Field[3] | Place; Object; Place + object |

*ANOVA[10]*: Analysis of Variance; tests for differences in firing rates across multiple spatial bins or conditions. *One-sided rank-sum test[11]*: firing rates were compared at nearby versus distant locations. *T-test[12]*: Firing rates were compared when the participant was in versus out of the place field. *OLS regression[13]*: Ordinary least-squares regression modeled spike counts using time, location, event type, and their pairwise interactions.

weaker-modulation neurons, prevalent both in humans and rats, offers new insight into the flexible coding strategies of the hippocampus and contributes to our understanding of spatial cognition across species.

## Results

In this study, we compare widespread place cell detection methods in rats and humans to understand how different analytical pipelines influence the identification of spatially tuned neurons, with the goal of informing future cross-species comparisons of hippocampal spatial coding (Fig 1A). We focus on two analytical approaches for detecting spatial tuning: spatial information (SI) and ANOVA F-statistics. Based on previous research, we evaluate SI using both fixed thresholding and permutation-based significance testing, while ANOVA is assessed through permutation testing, allowing us to examine how different classification strategies influence detection outcomes (Fig 1B). We systematically apply these detection pipelines to datasets from both humans and rats performing comparable linear track spatial navigation tasks to evaluate the extent to which rat and human place cells differ and to assess the role of methodological differences in contributing to these variations (Fig 1C, top and bottom). To further understand how these analytical methods relate to place field properties, we develop a simulation framework that varied parameters such as tuning width, firing rate, and trial-to-trial variability. Additionally, we estimate place cell features from rat, human, and simulated place fields and analyzed their structure in high-dimensional feature space using principal component analysis (PCA) to evaluate how spatial tuning metrics capture different place field properties and their interactions.

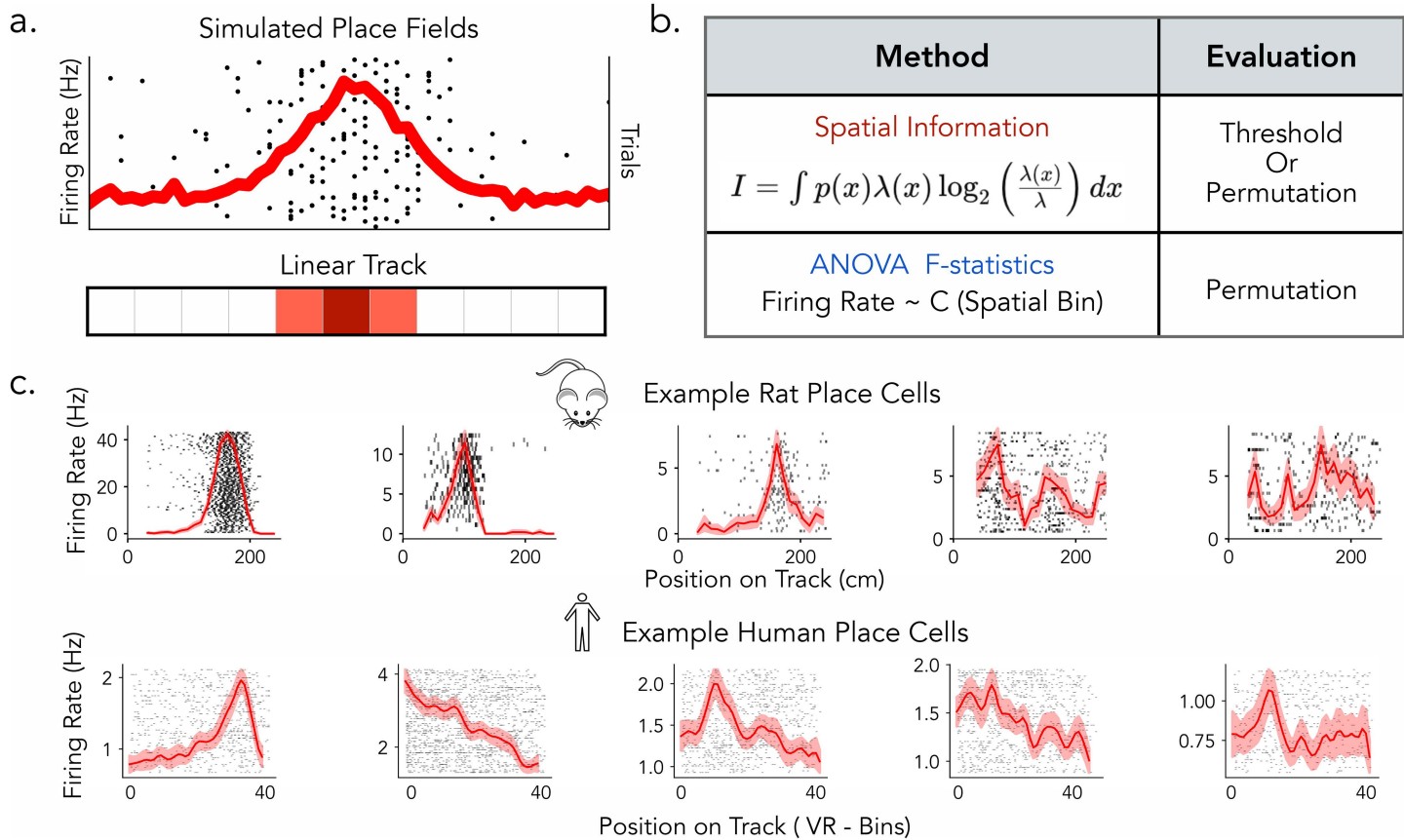

**Fig 1. Overview of place cells and detection methods in rats and humans.** a) Simulated place cells with varying spatial tuning properties on a linear track. b) Place cell detection pipeline commonly used in rats (Spatial Information) and in human (ANOVA). c) Examples of rat (top) and human (bottom) place cells, illustrating a range of tuning profiles from more prominent spatial selectivity (left) to weaker or more diffuse spatial modulation (right).

## Method-driven variability in rat place cell detection

We first evaluate these place cell detection methods by reanalyzing data from four Long-Evans rats that ran back and forth along a linear track while single-neuron activity was recorded (Fig 2A) [45]. We begin by comparing detection outcomes based on permutation testing (1,000 surrogates, $p < 0.05$) for both metrics. Overall and region-wise analyses reveals that permutation-corrected ANOVA identifies more spatially tuned neurons across hippocampal and entorhinal subfields than permutation-corrected spatial information (Fig 2B and **2C**). For ANOVA, the proportion of significant neurons increases in a sigmoidal pattern as a function of the F-statistic. This sharp transition around the significance threshold suggests high consistency between the raw score and permutation testing outcome (Fig 2D). In contrast, the proportion of neurons classified as significant by SI-based permutation testing increases more gradually with spatial information scores, suggesting weaker alignment between raw scores and surrogate testing (Figs 2E and S1A). While permutation-corrected results for ANOVA and spatial information show substantial agreement, they do not fully converge. As a function of the F-statistic, agreement between the two spatial tuning metrics was high at both low and high F-statistic but dropped near the ANOVA cutoff ($F = 2$) before recovering, indicating that disagreement was concentrated around the cutoff where classification is most sensitive to the choice of metric (Fig 2F). By contrast, agreement across SI values remained consistently high with only a modest upward trend and no clear dip

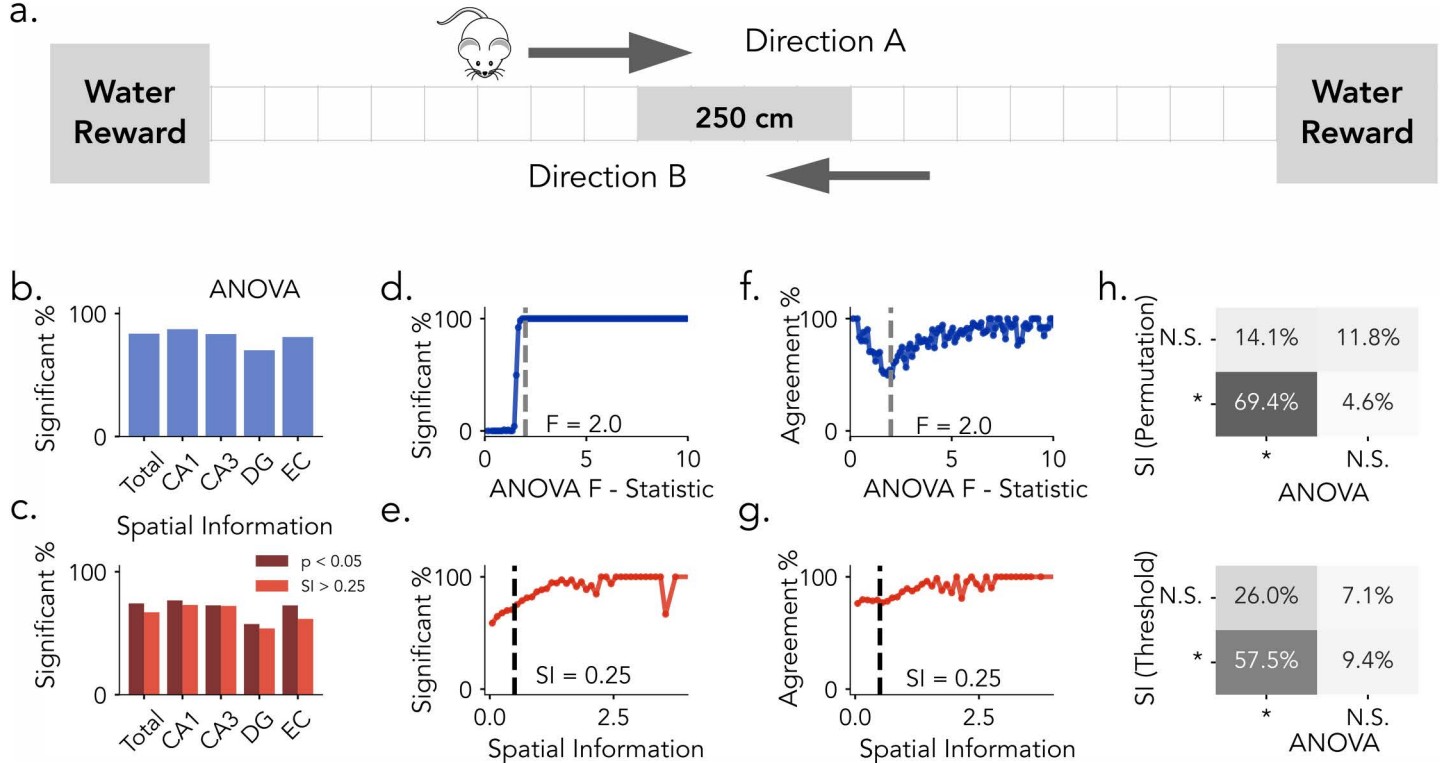

**Fig 2. Place Cell Identification Methods in Rats. a)** Task Schematic. Rats traverse bi-directionally along a 250 cm linear track to obtain water rewards located at each end (Directions A and **B**). **b)** Proportion of neurons identified as significant by ANOVA permutation testing across brain regions. CA1 and CA3: Hippocampus, DG: Dentate Gyrus, EC: Entorhinal Cortex. **c)** Proportion of neurons exceeding the SI threshold of 0.25 (light red) and those confirmed by SI permutation testing (dark red) across brain regions. **d)** Proportion of neurons classified as significant as a function of increasing ANOVA F-statistics. **e)** Same as in **d)**, plotted against Spatial Information Scores. **f)** Agreement between permutation-based SI classification and ANOVA classification as a function of increasing F-statistics. **g)** Same as **f.**, plotted against spatial information scores. Dashed lines in d. and f. mark the observed threshold for ANOVA (F = 2.0); dashed lines in e. and g. indicate spatial information threshold (SI = 0.25). **h)** Top: Comparison between SI permutation testing ($p < 0.05$) and ANOVA permutation testing ($p < 0.05$). Bottom: Comparison between SI thresholding ($SI > 0.25$) and ANOVA classification. Percentages indicate the proportion of neurons in each classification category.

at the threshold (SI = 0.25), suggesting that discrepancies are not localized to the SI cutoff (Fig 2G). Here, we use a 0.25-bit spatial information threshold, as applied in prior rodent studies, to provide a more liberal inclusion criterion. This allows us to capture a broader range of spatial tuning profiles, particularly in human data, where only 0.34% of units exceed a 0.5-bit threshold, compared to 10.1% exceeding a 0.25-bit threshold. This consideration is important for cross-species comparisons [24]. When comparing permutation testing on both spatial tuning metrics, a majority of neurons (69.4%) were jointly identified, but notable subsets were uniquely classified by ANOVA permutation testing (14.1%) or SI permutation testing (4.6%) (Fig 2H, top). Divergences became more pronounced when comparing ANOVA permutation testing with SI thresholding: only 57.5% of neurons were jointly identified, while substantial subsets were uniquely classified by ANOVA (26.0%) or by SI thresholding alone (9.4%) (Fig 2H, bottom). Together, these results show that ANOVA yields stronger alignment between raw scores and permutation testing and consistently identifies a larger population of spatially tuned neurons than spatial information, highlighting the impact of methodological choices on place cell classification.

## Effects of detection methods on human place cells

Building on our initial analyses in rodents, we next examine how place cell detection methods influence classification outcomes in humans. We reanalyze single-neuron recordings from patients with drug-resistant epilepsy using a dataset originally collected to study hippocampal and entorhinal function [14,46]. Participants performed a virtual navigation task in which they were passively moved along a linear track and instructed to learn and recall the locations of four objects (Fig 3A). The task closely resembles linear track paradigms used in the rat dataset, supporting method comparisons under comparable behavioral conditions. Across brain regions, permutation testing on both spatial tuning metrics (ANOVA and SI) identified smaller proportions of human place cells, and SI thresholding similarly yielded reduced proportions (Fig 3B and 3C).

Consistent with the rat findings, when comparing permutation-based significance testing to raw F-statistics, the proportion of neurons classified as significant by ANOVA exhibits a sigmoidal relationship with the F-statistic values (Fig 3E). On the other hand, compared to rats, SI-based classification in humans displayed greater variability and more pronounced divergence between fixed-threshold and permutation-based results (Figs 3F and S1B). We next assessed

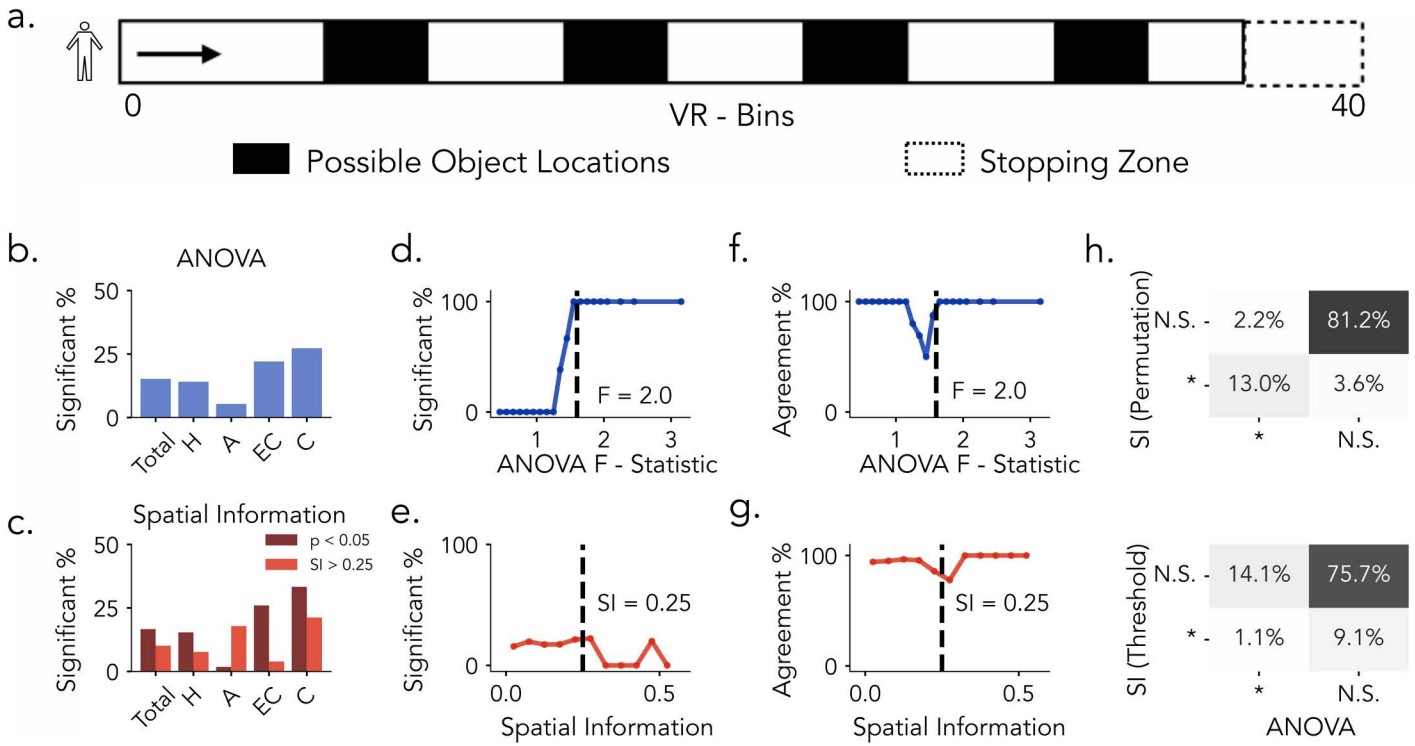

**Fig 3. Place Cell Identification Methods in Humans. a)** Task Schematic. Subjects are moved along a linear track (40 virtual units) to learn and recall the location of four objects. **b)** Proportion of neurons identified as significant by ANOVA permutation testing across brain regions. H: Hippocampus, A: Amygdala, EC: Entorhinal Cortex, C: Cingulate. **c)** Proportion of neurons exceeding the SI threshold of 0.25 (light red) and those confirmed by SI permutation testing (dark red) across brain regions. **d)** Proportion of neurons classified as significant as a function of increasing ANOVA F-statistic. **e)** Same as in **d)**, plotted against Spatial Information Scores. **f.** Agreement between permutation-based SI classification and ANOVA classification as a function of increasing F-statistic. **g)** Same as **f.**, plotted against spatial information scores. Dashed lines in **d.** and **e.** mark the observed threshold for ANOVA ($F = 1.6$); dashed lines in **f.** and **g.** indicate the spatial information threshold ($SI = 0.25$). **h)** Top: Comparison between SI permutation testing ($p < 0.05$) and ANOVA permutation testing ($p < 0.05$). Bottom: Comparison between SI thresholding ($SI > 0.25$) and ANOVA classification. Percentages indicate the proportion of neurons in each classification category.

the correspondence between the two spatial tuning metrics under permutation-based evaluation. When examined as a function of the ANOVA F-statistic (Fig 3F), agreement between ANOVA- and SI-based classifications was generally high at both low and high values but exhibited a pronounced dip around the ANOVA significance boundary ($F = 2$). This pattern indicates that neurons within this range of F-statistics are differentially classified by the two approaches, reflecting heightened sensitivity to the choice of spatial tuning metric. When examined as a function of spatial information (Fig 3G), agreement remained comparatively stable across values, with only a modest decline near the SI threshold (0.25), suggesting that discrepancies are less localized and less pronounced than those observed for ANOVA. Notably, these trends closely parallel the results observed in rats, highlighting consistent cross-species differences in how ANOVA and SI capture neurons with weak or intermediate levels of spatial tuning. In general, comparing permutation-based significance testing across spatial tuning metric, we observe overlap between methods: 12.1% of neurons are jointly classified as spatially tuned, while 3.4% and 2% are uniquely identified by SI and ANOVA, respectively (Fig 3B, top). In contrast, applying a fixed threshold to SI scores ($SI > 0.25$) led to substantial divergence,only 1% of neurons are jointly classified, while 9.1% and 13.1% are uniquely identified by SI and ANOVA, respectively (Fig 3H, bottom). In summary, both permutation-based methods (ANOVA and SI) as well as fixed SI thresholding identified fewer human neurons as place cells compared to rats. Similarly in rats, ANOVA showed strong concordance between raw F-statistics and permutation testing, whereas SI exhibited weaker alignment between raw and surrogate evaluations.

## Effects of firing-rate map construction parameters

Prior methodological work has shown that the parameters used to construct firing-rate maps, such as bin size and smoothing kernel, are themselves a source of variability in spatial analyses [47]. Consistent with their findings, we demonstrate that methodological choices in rate-map construction, including smoothing and spatial bin resolution, yield systematic, metric-specific effects that ultimately determine which neurons are classified as spatially tuned. Specifically, we find that applying a smoothing kernel inflates ANOVA F-statistics while deflating spatial information scores (S2 Fig). Additionally, using higher spatial bin resolutions increases spatial information scores but decreases ANOVA F-statistic values (S3A–L Fig), making fixed-threshold methods more dependent on analysis parameters such as bin size and smoothing kernel compared to permutation-based approaches, which remain more robust across varying analysis settings. (S3M-N Fig). These differences highlight that spatial information thresholding is sensitive to analysis parameters such as smoothing and spatial binning. Overall, these findings indicate that fixed thresholding methods are highly dependent on analysis parameters and may classify different subsets of neurons under varying conditions, whereas permutation-based significance testing provides a more stable and robust approach for detecting spatial tuning and shows greater consistency across species and analytical methods.

## Statistical distributions of spatial tuning metrics differ across species

To characterize variation in spatial tuning metrics across species, we examined the distributions of spatial information (SI) scores and ANOVA F-statistics for rat and human single-neuron datasets, respectively (Fig 4). Statistical significance for each metric was assessed independently using permutation testing (1,000 surrogates; $p < 0.05$). For rat neurons, ANOVA F-statistics ranged from 0.14 to 400.12, and SI scores from 0.01 to 4.54 (Fig 4A). A substantial proportion of neurons met significance criteria under both SI and ANOVA, with additional neurons identified exclusively by ANOVA. Several neurons exceeded the permutation-based SI threshold but fell below the commonly used fixed cutoff of $SI > 0.25$, highlighting divergence between thresholded and surrogated detection criteria. In comparison, human neurons exhibited a narrower range of tuning metrics, with ANOVA F-statistics from 0.43 to 3.12 and SI scores from 0.00 to 3.64 (Fig 4B). Fewer neurons reached significance under either metric, and there was reduced overlap between SI- and ANOVA-identified neurons relative to rodents. Fixed SI cutoffs were more limiting in this context, while ANOVA identified a broader set of neurons across varying tuning strengths. We observed that a subset of rodent neurons exhibited SI and F-statistic values within the range

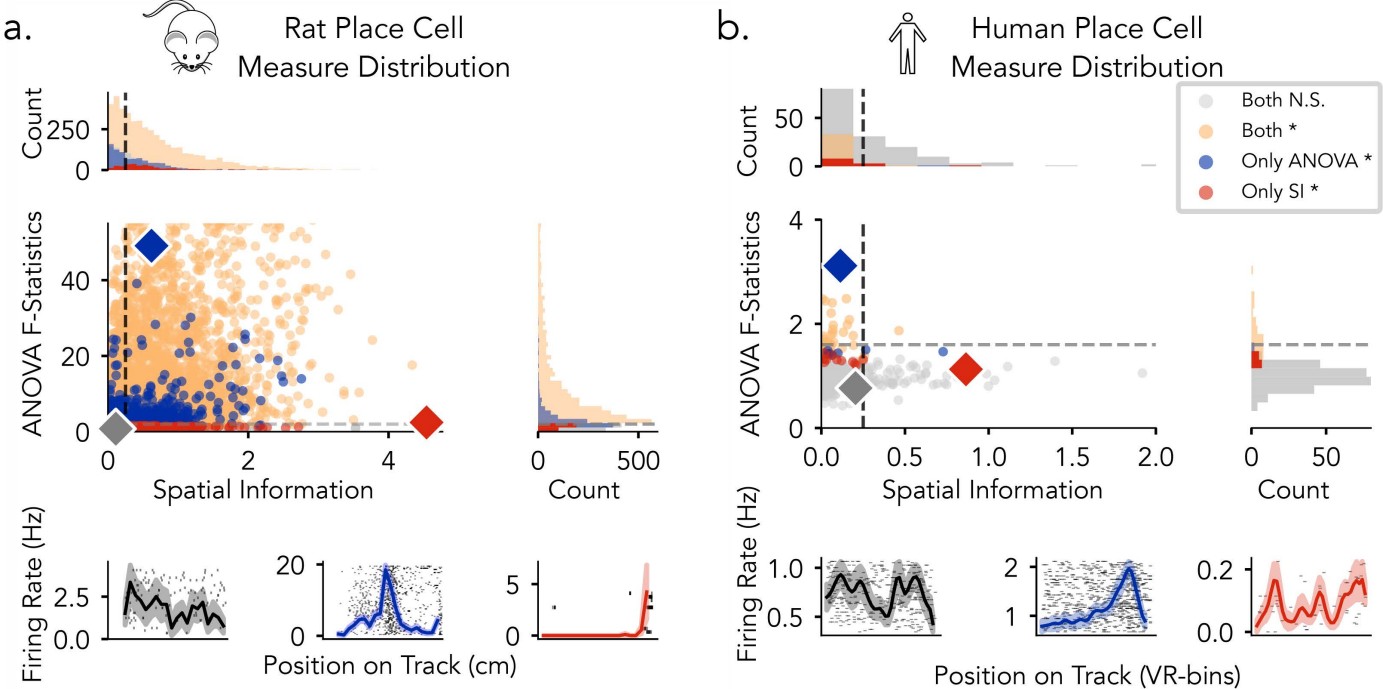

**Fig 4. Statistical distribution of place cell measures in rats and humans. a)** Rats: ANOVA F-statistics 0.14–400.20; spatial information 0.01–4.54. **b)** Humans: ANOVA F-statistics 0.43–3.12; spatial information 0.01–3.64. Neurons are plotted by spatial information (x-axis) and ANOVA F-statistic (y-axis). Each point represents a single neuron. Dashed lines indicate thresholds (SI = 0.25; F = 2.0 for rats, 1.6 for humans). Marginal histograms show metric distributions and neuron counts by category. Bottom, Example firing rate maps for gray (low SI, low ANOVA), blue (high ANOVA), and red (high SI).

observed in the human data, indicating overlap in tuning profiles across species. Notably, while rat neurons spanned a substantially broader range of SI and F-statistic values, their distributions overlapped with human neurons at the lower end, indicating shared profiles of weak or intermediate spatial tuning across species, which may subserve comparable functional roles.

## Modeling the influence of place cell properties on detection statistics

The analyses in the preceding sections reveal consistent differences between spatial information (SI) and ANOVA in evaluating the spatial tuning of place cells. However, it remains unclear how specific place cell properties influence these spatial tuning metrics. In empirical datasets, such effects are difficult to isolate because the ground-truth spatial tuning properties of individual neurons are unknown. To address this limitation, we developed a simulation framework that generates synthetic tuning curves with precisely controlled parameters, providing ground-truth benchmarks for systematically assessing how different detection metrics respond to variations in tuning properties (Fig 5). Specifically, this approach allows us to independently vary features such as peak firing rate, field width, background firing, noise, place field consistency across trials, and presence ratio, while holding other variables constant to isolate the effect of each property on detection metrics. Modeling at the level of tuning curves offers a controlled and interpretable approach for assessing how spatial information and ANOVA F-statistics respond to variations in place cell activity.

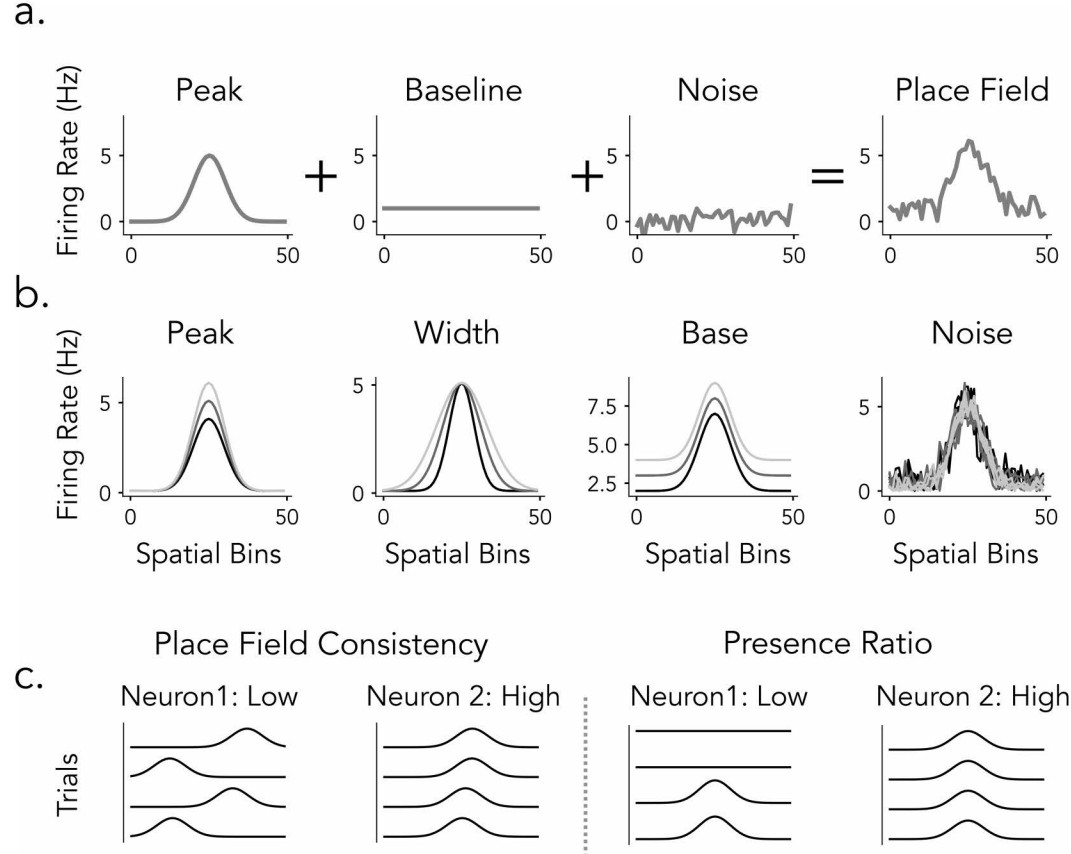

**Fig 5. Place Field Simulations. a)** Place Field Components. Simulated place fields are generated by combining three components: Peak - Gaussian-shaped firing rate profile, Baseline - constant firing rate across space, and Noise - random fluctuations in firing rate. The final place field (right) is produced by summing these components. **b)** Field Properties. Place field tuning parameters are varied across neurons to model population-level diversity. Peak - modulation of peak firing rate, Base - adjustment of background firing rate, and Width - spread of the place field, define each neuron's trial-averaged spatial tuning profile. In contrast, Noise - random fluctuations added to the firing rate, varies across trials, introducing realistic trial-to-trial variability in field expression. **c)** Trial-Level Field Consistency. To assess within-neuron reliability, we compute two metrics: Place Field Consistency (left) quantifies the spatial stability of peak firing across trials, while Presence Ratio (right) measures the fraction of trials in which a detectable place field is expressed, reflecting tuning persistence over time.

## Peak firing rate enhances spatial tuning

We examined how variation in peak firing rate (1–20 Hz), under biologically plausible conditions and with all other parameters held constant, affects spatial tuning detection metrics (Fig 6A). As the peak firing rate increased, the spatial information score rose because it is sensitive to how much more the neuron fires in certain locations compared to others—i.e., the contrast between peak and average firing rates (Fig 6B). At the same time, ANOVA F-statistics also increased because the firing rates became more different across spatial bins (i.e., locations), which increases the variance between spatial bins (Fig. 6B). Importantly, the variability within each bin (i.e., trial-to-trial variability) was held constant, so the increase in the F-statistic reflects more structured, spatially organized firing rather than noise. These results indicate that both metrics are sensitive to peak firing rate, but through distinct mechanisms: spatial information quantifies the contrast between a neuron's peak firing rate and its overall mean rate across spatial locations, whereas ANOVA assesses the proportion of firing rate variance systematically explained by spatial position relative to variation across trials (Fig 6C).

## Variation in place field width exhibits non-monotonic effects

We next investigated how spatial tuning width, defined as the standard deviation ($\sigma$) of a Gaussian place field, varied from 1 to 20 spatial bins out of a total of 50, influences spatial tuning detection while holding all other parameters constant (Fig 6A). Both spatial information and ANOVA F-statistics exhibited non-monotonic dependencies on field width (Fig 6B). Spatial information peaked at narrow widths ($\sigma \approx$ 2-3 bins), where sharply localized firing created strong contrast between active and inactive locations. In contrast, ANOVA F-statistics were maximized at broader widths ($\sigma \approx$ 6-8 bins), reflecting an optimal balance between structured variation across spatial bins and maintained trial-level variability. At larger field widths, both metrics declined: spatial information decreased as spatial contrast diminished, while ANOVA F-statistics dropped as the tuning profile became increasingly flat and spatial variance was reduced. Accordingly, the joint metric analysis (Fig 6C) revealed a curved relationship, with both metrics elevated only within a restricted range of intermediate widths. This finding indicates that similar detection scores can emerge from distinct underlying spatial profiles.

## Differential sensitivity to baseline firing rate

To assess the impact of background activity on detection sensitivity, we manipulated the baseline firing rate from .5 to 5 Hz, while keeping all other parameters fixed. Increasing baseline elevated overall firing rates uniformly across trials and spatial field without altering the underlying tuning shape (Fig 6A). This manipulation had a marked effect on spatial information scores (Fig 6B, red) which decreased steadily as baseline rose. Since spatial information is sensitive to the ratio of peak to average firing, higher background activity compresses this contrast and diminishes the score. ANOVA

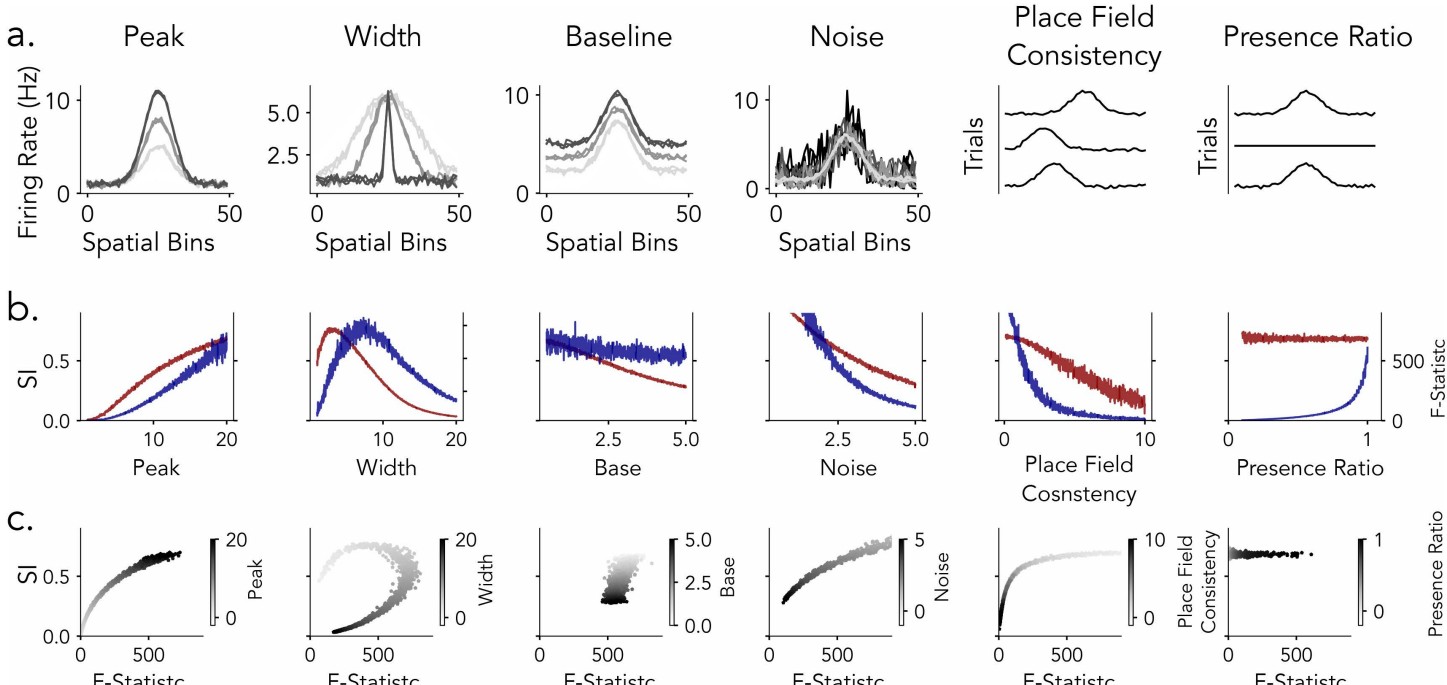

**Fig 6. Impact of place field features on spatial information and ANOVA statistics. a)** Simulated firing rate maps illustrating variation across six place field parameters: peak firing rate, width, baseline firing rate, noise, place field consistency, and presence ratio. **b)** Spatial information (SI, red) and ANOVA F-statistics (blue) as a function of each place field parameter. **c)** Joint distributions of SI (y-axis) and ANOVA F-statistics (x-axis), with grayscale indicating values of the corresponding parameter.

F-statistics (Fig 6B, blue), by contrast, exhibited an initial drop at low baseline values but remained relatively stable across the remainder of the range. Since ANOVA captures variance across spatial bins relative to trial-to-trial variability, uniform increases in baseline preserve the spatial structure of the tuning curve and exert a smaller influence on the metric (Fig 6C, blue). We observed a similar trend in empirical recordings from putative place cells in human and rodents. Human neurons, which typically show elevated baseline activity, had lower SI scores despite consistent spatial tuning, whereas the example rodent neurons with sparse firing showed high spatial information score without reliable tuning. These findings highlight that SI may overestimate tuning in sparse neurons and underestimate it in neurons with high background activity - patterns that are particularly common in human recordings, where place fields tend to be less sharply localized and background firing rates are often high.

### ANOVA is more responsive to trial level consistency than spatial information

To evaluate how trial-level variability affects detection metrics, we manipulated three aspects of tuning stability: additive noise, trial-wise place field shifts, and the presence ratio (Fig 6A). In all cases, the trial-averaged spatial tuning profile was preserved, while trial-to-trial reliability was systematically degraded.

For additive noise (Fig 6A), independent Gaussian noise with increasing standard deviation (.5–5 Hz) was added to each trial. Both metrics declined as noise increased, but ANOVA F-statistics dropped sharply due to rising trial-level variance relative to across-bin differences. Spatial information decreased more gradually, remaining moderately high even at large noise levels due to its reliance on trial-averaged firing (Fig 6B).

For place field shifts (Fig 6A), the location of each trial's tuning curve was jittered by a random offset (0–5 spatial bins). ANOVA F-statistics decreased rapidly with increasing shift magnitude, reflecting diminished spatial alignment across trials. Spatial information, by contrast, remained relatively stable, again reflecting averaging across spatial locations (Fig 6B).

For presence ratio manipulations (Fig 6A), place fields were selectively silenced on a subset of trials, reducing their presence ratio from 1.0 to 0.1. ANOVA values increased steeply with higher presence ratios, indicating sensitivity to consistent expression of tuning. Spatial information, however, remained largely flat, failing to capture trial-wise field dropout (Fig 6B).

Metric–metric comparisons (Fig 6C) revealed consistent dissociations across all three manipulations: spatial information scores stayed relatively high even when ANOVA values were low, especially under high noise, large shifts, or sparse field expression. These results demonstrate that spatial information is largely insensitive to trial-to-trial fluctuations, while ANOVA more directly captures trial-level consistency.

Together, these simulations demonstrate that spatial information and ANOVA F-statistics capture distinct, complementary aspects of spatial tuning in single neurons. Spatial information is most sensitive to sharp, high-contrast firing fields and is strongly modulated by the ratio of peak to baseline activity, but is relatively insensitive to trial-to-trial instability and field dropout. In practice, however, this limitation is often mitigated in rodent place-cell studies by assessing stability through additional analyses which evaluate the reliability of place-related activity by examining rate maps across different groups of trials (e.g., split halves or even-odd splits) to ensure that identified fields are consistent across trials [35,40,48]. On the other hand, ANOVA is responsive to structured spatial variance, which is consistent, and reliably expressed tuning across trials. Notably, both metrics exhibited non-monotonic dependencies on place field width, with maximal detection sensitivity occurring at intermediate tuning widths, underscoring the importance of field size in shaping metric outcomes. Our modeling further reveals that the raw scores of each metric are differentially sensitive to specific place field properties, indicating that variations in field characteristics can lead to divergent detection outcomes depending on the metric employed.

### Place cell feature estimates

We next estimated several features of place cell activity from rat, human, and simulated neurons, including peak firing rate, average firing rate, the peak-to-average rate ratio, place field width, number of place fields, presence ratio, and

even-odd correlation (S4 and S5 Figs). Consistent with our simulation findings, we found that across datasets, SI correlated most strongly with the peak-to-average firing-rate ratio, whereas ANOVA correlated more with even–odd reliability, underscoring that the two metrics capture different aspects of tuning properties (S4 and S5 Figs).

### PCA of tuning features reveals divergent structure for spatial information and ANOVA

Having established how individual properties correlate with spatial tuning metrics, we next sought to examine how interactions among multiple place field features jointly influence spatial tuning metrics. Principal component analysis (PCA) was applied to characterize the low-dimensional structure of feature interactions and to identify dominant patterns underlying spatial tuning across datasets.

PCA of rat hippocampal neurons revealed that gradients in ANOVA F-statistics and Spatial Information score occupy distinct, though overlapping, axes in feature space (Fig 7A and 7B). This organization is clarified by the feature loading vectors (Fig 7C): the SI gradient aligns most strongly with peak-over-average firing rate (red) and runs opposite to place field width (magenta), reflecting SI's sensitivity to firing contrast and field sharpness. In contrast, the F-statistic gradient is oriented along even–odd correlation (yellow) and place field consistency (orange), and is negatively associated with the number of place fields (dark red), highlighting ANOVA's emphasis on trial-to-trial reliability. The ordering and orientation of these loadings indicate that SI and ANOVA capture distinct combinations of underlying place field properties.

These differences are further illustrated by example firing rate maps sampled from each distribution (bottom panels): neurons with high SI values exhibit sharply localized, high-contrast tuning profiles, while low SI neurons show flatter responses. High F-statistic neurons display strong trial-to-trial consistency in their spatial responses, whereas low F-statistic neurons exhibit irregular or noisy activity across bins.

Feature loadings and their ordering in humans and simulations were comparable to those in rats, with consistent contributions from key features such as even–odd correlation, peak-over-average firing rate, and place field width. Moreover, the principal axes defined by these features correspond to gradients in ANOVA F-statistics and spatial information (Figs 7D, 7E, S6, and S7). Together, these results demonstrate that SI and ANOVA emphasize distinct axes of the neural feature space. SI is more influenced by tuning sharpness and signal contrast, while ANOVA is more closely aligned with trial-level stability. PCA thus reveals that these two commonly used spatial tuning metrics occupy partially overlapping but separable subspaces, highlighting their complementary roles in characterizing spatial coding in the hippocampus across species and in simulations.

## Discussion

### Overview and significance of the findings

Cross-species comparisons are critical for uncovering core computational principles of hippocampal function that are either conserved or divergent across biological systems. Despite rich literatures on place cells in rodents and humans, few studies have directly compared them under comparable behavioral paradigms and analytic frameworks [1,10,21,22]. Methodological conventions differ across species – rodent researchers tend to use spatial information (SI)-based methods in conjunction with other measures including place field stability (Table 1), whereas human researchers more often employ ANOVA-based approaches (Table 2). This lack of alignment makes it difficult to determine whether observed differences reflect true biological variation, task-related factors, or methodological differences. To begin to fill this gap, this study addresses an important methodological and conceptual challenge in cross-species spatial coding: how detection metrics influence the classification and interpretation of place cells. In this study, we systematically evaluate two spatial tuning metrics: SI and ANOVA, across human and rat single-neuron datasets, applying both fixed thresholds and permutation based significance tests to assess how different classification criteria influence place cell detection. To assess how spatial tuning metrics respond to place-field features, we used a simulation framework with systematically varied parameters and known ground truth.

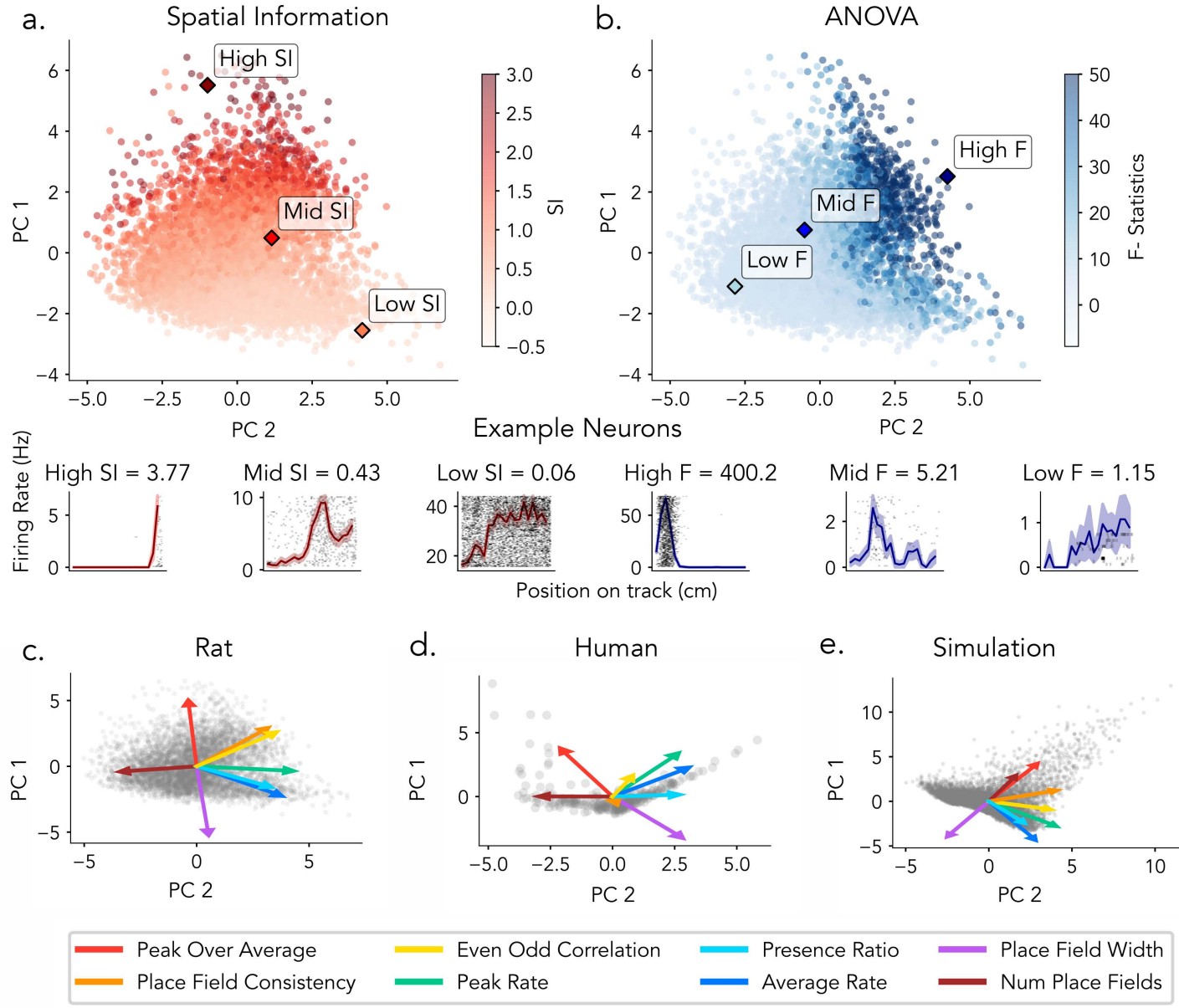

**Fig 7. Estimated tuning features reveal distinct structures captured by SI and ANOVA.** Principal component analysis (PCA) was applied to tuning features derived from hippocampal neurons, including: even–odd correlation, peak-to-average ratio, place field width, presence ratio, place field consistency, number of place fields, average firing rate, and peak firing rate. **a)** Top: Rat neurons projected onto the first two principal components, colored by spatial information (SI). Bottom: Example rat neurons with high, median, and low SI, shown with corresponding firing rate maps. **b)** Top: Same PCA projection as in a, but colored by ANOVA F-statistic. Bottom: Example rat neurons with high, median, and low F-statistics, shown with corresponding firing rate maps. **(c–e)** Principal component analysis (PCA) projections and loading vectors of place cell features for rats **(c)**, humans **(d)**, and simulations **(e)**. Points represent neurons projected into the first two principal components. Arrows indicate the loading vectors for each feature, with their direction and length representing the contribution and magnitude of each feature to the first two principal components.

Our findings show that spatial tuning metrics are differentially responsive to place cell features, and that both the choice of metric and classification criterion (permutation-based vs. thresholding) directly influence which neurons are classified as spatially tuned. Cross-species comparisons revealed that rat place cells span a broad spectrum of spatial tuning scores, ranging from sharply defined, high-contrast fields with elevated SI and ANOVA values to broad but consistent fields with lower scores on both metrics. In contrast, human place cells often exhibit broader and more consistent tuning, with a tendency to cluster in the low SI/low ANOVA quadrant that overlaps with the lower range of the rat distribution. These observed differences may also reflect the fact that human and rodent datasets are collected under different experimental conditions, including passive versus active navigation, real-world versus virtual environments, as well as differences in anatomical sampling, rather than purely intrinsic species traits [7,34,27, 49–53]. Such cross-species variation in tuning properties helps contextualize prevailing methodological choices. In rodents, the relative abundance of sharply localized, high-contrast place fields has likely contributed to the widespread use of SI-based approaches, whereas in humans, the broader tuning, often accompanied by conjunctive coding of spatial and non-spatial variables, appears more readily detected with ANOVA. Our results suggest that methodological choices shape place-field classification and interpretation across species, providing a foundation for more rigorous comparative research on spatial coding.

## Methodological findings and recommendations

### Spatial tuning metric

Our analyses indicate that spatial information scores and ANOVA F-statistics emphasize different place-field features. SI primarily reflects the peak-to-average firing rate ratio: neurons with sharply localized, high-contrast fields attain high SI values, whereas neurons with elevated baseline activity more common in human recordings exhibit lower scores. SI is less responsive to trial-level variability because it is computed from the average firing rate across trials. Prior rodent studies often address this issue through additional stability measures [25,40,42]. Further, researchers could consider using bootstrapping to quantify how sensitive the SI estimate is to trial-to-trial variability, providing an indirect assessment of firing-field stability [54]. By contrast, ANOVA F-statistics measure how robustly a neuron's firing varies with position across repeated traversals. Formally, the F-statistic is the ratio of variance in firing rates between spatial locations to the residual variance across trials within each location. High values arise when positional differences are pronounced and trial-level responses are internally consistent. In summary, SI primarily captures field contrast whereas ANOVA F-statistics evaluate spatial differentiation and trial-level reliability, providing complementary views on spatial coding. Neurons with sharply localized firing fields may achieve high spatial information scores even when their firing is highly variable across trials due to task-dependent modulation [55]. Neurons with complex, broader yet trial-consistent spatial firing patterns, as commonly observed in the rodent ventral hippocampus and in human single-neuron recordings [14,34], are more likely to be identified by ANOVA-based criteria.

### Place cell detection pipelines

Place cell detection methods often rely on fixed thresholding of spatial tuning metrics and/or statistical testing [10,11,14,24–27,29,36,53]. While threshold-based classification remains widely used, these metrics are sensitive to quantitative analysis choices, such as spatial resolution (e.g., bin size) and smoothing kernel width (S2 and S3 Figs). These parameters shape the distributions of raw scores and may introduce instability when fixed thresholds are applied across datasets with different firing statistics or behavioral structures. Permutation-based significance testing offers an alternative approach, generating a null distribution for each neuron through within-session shuffling. By adjusting for neuron-specific firing variability and session-level statistics, this method tends to yield more consistent classification outcomes across different analysis settings compared to fixed-threshold approaches (S3M-N Fig).

## In relation to other methods

In addition to SI and ANOVA, methods such as generalized linear models (GLMs) and information-theoretic measures provide alternative ways to characterize and compare spatial tuning. GLMs, conceptually similar to ANOVA, allow firing rates to be modeled as a function of multiple behavioral covariates, providing a flexible approach for handling correlated predictors, though they typically require larger datasets and more careful regularization than is available in most current human single-neuron recordings [56]. Kullback-Leibler based measures quantify differences in firing-rate or population-activity distributions across task or environmental conditions [57,58] and are better suited for assessing changes in the shape of firing distributions. These approaches illustrate the range of available methods for assessing spatial modulation across different datasets and analytical goals.

## Extension to 2D environments and open-field foraging

Both SI and ANOVA have been used in two-dimensional navigation tasks, with SI widely applied in rodent open-field studies [7,25,26,30,39] and ANOVA used in 2D human navigation tasks [10,11]. We verified that the comparative properties of SI and ANOVA extend to two-dimensional environments using simulated 2D place fields (S8 Fig). These simulations have consistent results with the 1D simulations, indicating that the contrast-versus-consistency distinction we identify is not specific to linear tracks but reflects general properties of the metrics themselves. This result is expected, since both SI and ANOVA operate on activity across bins, and do not inherently differentiate such patterns across 1 or 2 (or more) dimensions. Factors that influence these metrics in 1D, such as baseline firing rates, field broadness, and variability across traversals, affect the metrics analogously in 2D (S8 Fig). Together, these observations indicate that the comparative properties of SI and ANOVA described in this study are applicable across linear tracks, and 2D environments.

## Recommendations

Together, our findings support a set of best practices for reliable and comparable place cell detection. First, while both spatial information and ANOVA effectively identify strongly tuned place cells, they differ in their treatment of borderline or noisy cases. Spatial information tends to highlight neurons with high-contrast, sharply tuned firing fields, whereas ANOVA is more likely to detect neurons with weaker but consistent modulation across trials, often characteristic of the tuning profiles observed in human recordings. Thus, the key distinction between these metrics lies not in their ability to detect canonical place cells, but in how they classify ambiguous neurons near the threshold of significance. Studies should consider the goals of their analyses and the likely properties of the place cells in their dataset to choose appropriate methods. Second, permutation-based significance testing provides greater robustness than fixed thresholds by generating neuron-specific null distributions that account for baseline variability and reduce dependence on analysis parameters such as bin size and smoothing. Together, these practices offer a more principled and reproducible framework for identifying spatially tuned neurons and enable more accurate comparisons of hippocampal spatial coding across species.

## Relevance to cross-species explorations

Our analyses revealed both differences and meaningful overlaps in the statistical distribution of spatial tuning across species. Rat neurons exhibited a broad dynamic range on both the SI and ANOVA axes, including sharply tuned cells with high spatial contrast as well as more broadly and weakly modulated neurons. Human neurons, by contrast, clustered within a narrower range and were largely absent from the extreme high-SI and high-ANOVA region that characterizes the upper end of the rodent distribution. However, these cross-species differences may also reflect characteristics of the human dataset, which, as is normal for human datasets, is smaller than can be recorded from animal models and is obtained from patients and under constraints of recording in a hospital setting. Factors such as restricted anatomical coverage, medication status, reliance on VR-based navigation, and variability in patient engagement can influence the

measurable range of spatial tuning in human place cells. Thus, the narrower dynamic range observed in the human data may be shaped by these experimental and clinical factors in addition to, rather than solely by, species-level differences. Nevertheless, subsets of neurons from both species fell within overlapping SI–ANOVA ranges, indicating that neurons in both species can exhibit comparable spatial tuning profiles, even if their overall prevalence and sharpness differ. Notably, permutation-based ANOVA was more effective than fixed SI thresholds at identifying these overlapping neurons.

Importantly, these moderately tuned neurons may still play a important functional role. In rodents, similar broad tuning profiles are more commonly found in the ventral hippocampus, a region associated with spatial and non-spatial contextual coding [7,34]. In humans, spatially modulated responses are typically examined in behavioral paradigms that also engage episodic memory, decision-making, and goal-directed behavior, so that recorded neurons may reflect integrated, conjunctive coding schemes rather than purely spatial tuning [10,12,14]. These findings emphasize that moderate spatial tuning is not necessarily indicative of noise or weak signal, but may reflect meaningful coding strategies adapted to the task demands of each species. Considering these potential ecological and task-related differences is important for interpreting hippocampal function across species and for building cross-species models of spatial representation.

Beyond the difference in species, there are several additional differences between the empirical datasets compared here, which are also potentially relevant to understanding the results. Notably these differences are not specific to these particular datasets, but reflect typical differences in experiments between rodents and humans. Here we briefly note these differences and how they may relate to the methodological findings and suggest them as avenues for future work comparing between species.

### Anatomical differences

Anatomical sampling differences likely contribute to the observed variation in spatial coding between rodents and humans. In rodents, place field size and spatial precision vary systematically along the dorsoventral axis, with small, sharply tuned fields in dorsal hippocampus and larger, more diffuse fields in ventral regions [7,34]. Human intracranial recordings are constrained by clinical needs, leading to limited and variable anatomical coverage. While the rodent dorsoventral axis is often proposed to correspond functionally to the human anterior-posterior axis, it remains unclear whether human place cells follow a similar gradient [51,52].

### Virtual environments vs. Real world

Another important consideration when interpreting cross-species differences in place cell properties is the use of virtual environments in human experiments versus real-world navigation in most rodent studies. In rodents, previous work has shown that virtual reality (VR) leads to broader place fields, decreased stability, reduced spatial selectivity, and lower firing rates compared to freely moving conditions [49,59]. By contrast, human intracranial studies are typically conducted in desktop-based 'video game–like' virtual environments due to practical and clinical constraints. As a result, some of the observed differences in place field properties across species may not solely reflect biological variation, but may also be shaped by differences in recording context. Future research using more immersive or naturalistic paradigms in humans may help clarify the extent to which VR-specific factors influence hippocampal spatial coding.

### Passive vs. Active navigation

A related factor influencing the interpretation of place cell across-species is the distinction between passive and active navigation. Previous rodent work demonstrates that removing self-motion cues, such as through passive transport or head-fixed conditions, can weaken or destabilize spatial tuning and that visual input alone is insufficient to sustain robust place-cell firing during passive viewing [50,53,27]. These findings suggest that the passive nature of the human navigation paradigm may contribute to the smaller proportion and broader tuning of spatially modulated neurons observed here.

Future studies using actively controlled or more immersive navigation paradigms in humans will be important for determining the extent to which passive movement influences observed spatial coding patterns.

### Training, task engagement, and clinical constraints

Differences in training protocols, attentional demands, task engagement, and clinical context likely contribute to cross-species variability in observed place cell properties. Rodent experiments typically involve repeated exposure and reward-driven behavior, under which place-field properties stabilize and exhibit experience-dependent refinements, particularly when animals attend closely to spatial context [60–63]. Human intracranial recordings are conducted with epilepsy patients during clinical monitoring, where sessions are comparatively shorter and pre-training is limited. Factors such as attention, fatigue, alertness, and medication can vary considerably in this setting, potentially influencing place-field properties and spatial tuning in humans [64]. These contextual differences in training intensity, attentional state, and clinical constraints should therefore be considered when interpreting cross-species variability in hippocampal spatial representations.

### Future directions

In summary, our analyses indicate that methodological choices substantially influence place-cell detection. By introducing a statistically grounded detection framework, this work provides a foundation for disentangling methodological from biological variation and for developing a more generalizable account of spatial coding across species. Across species, both rats and humans contain overlapping populations of weakly tuned neurons that may contribute not only to spatial coding but also to broader representations of context, task structure, and memory. A key direction for future research is to determine whether these shared populations support comparable computational functions across species and, more broadly, whether humans exhibit the full spectrum of place-field properties observed in rodents. Addressing these questions will require accounting for differences in anatomical sampling, task design (e.g., virtual reality vs. real-world environments; passive vs. active navigation), training, and recording constraints. These efforts will be essential for studying cross-species principles of spatial coding.

## Methods

### Ethical statements

This study involved epilepsy patients who underwent surgical implantation of Behnke-Fried microelectrodes at four hospital sites: Emory University Hospital (Atlanta, GA), UT Southwestern Medical Center (Dallas, TX), Thomas Jefferson University Hospital (Philadelphia, PA), and Columbia University Medical Center (New York, NY). This study was approved by the Institutional Review Boards of Columbia University, Columbia University Medical Center, Emory University, University of Texas Southwestern, and Thomas Jefferson University, and were conducted in accordance with the principles of the Declaration of Helsinki. Written informed consent was obtained from all participants prior to their inclusion in the study.

### Literature search

To examine the existing literature on place cell analyses, we conducted a review combining automated and manual search strategies to identify studies of single-neuron recordings examining place cell activity across species, using the LISC Python tool [65] to support automated literature collection from the PubMed database. Our goal was to characterize and compare methodological approaches in both rodent and human studies (Table 1 and 2). For rodent research, which reflects a large literature, we sub-selected a sample of studies spanning a diversity of analytical pipelines. For human research, which is a much smaller literature, we included all published studies reporting intracranial single-unit recordings during (virtual) spatial or navigational tasks. Each paper was reviewed for key methodological features including recording site, behavioral task, spatial tuning analysis, and classification criteria.

## Datasets

### Rat dataset

We analyzed a publicly available dataset comprising 119 recording sessions from four male Long-Evans rats (250–400 g) navigating a 250 cm linear track for water rewards at each end [45,66]. Each animal was chronically implanted with silicon probes targeting the medial temporal lobe (MTL). Single-neuron activity was extracted from microwire recordings using KlustKwik for automated spike sorting, followed by manual curation with Klusters. Full details of the recording procedures and preprocessing steps are available in [45].

For spatial analyses, the track was divided into 26 evenly spaced bins. To reduce edge effects near reward sites, the first and last three bins were excluded. Quality control criteria were applied to an initial pool of 8,500 neurons to ensure data reliability. Sessions with fewer than 15 forward or backward trials were excluded, as were neurons with firing rates below 0.2 Hz or above 20 Hz, or total spike counts less than 50. Traversals were excluded if running duration deviated more than two standard deviations from the session mean. A minimum running speed threshold of 5 cm/s was enforced, and neurons with spike presence ratios below 50% were removed in the direction of interest. After applying these criteria, a total of 3,089 neurons were included for one running direction and 3,338 neurons for the opposite direction.

### Human dataset

We reanalyzed human single-neuron recordings from 306 neurons in the MTL region of 19 neurosurgical patients under-going intracranial monitoring for treatment of drug-resistant epilepsy from a previous experiment [14,46]. Neuronal activity was recorded while participants performed a virtual object–location memory task. Participants navigated a virtual linear track to learn object locations during encoding (2 trials per object), followed by a retrieval phase where they recalled the locations from memory (14 trials per object). Each session included 4 unique objects, for a total of 64 trials. For analysis, we excluded neurons with mean firing rates below 0.1 Hz or above 20 Hz to ensure stable and physiologically plausible neuron activity. For spatial analyses, the virtual track was linearly binned into 40 equally spaced segments, and neuronal firing rates were aligned to the participant's position along the track.

### Simulation framework

We developed a simulation framework to generate synthetic firing rates across spatially binned locations, with tunable place field properties, in order to systematically evaluate detection metrics in both 1D linear track and 2D environments. Notably, these simulations model trial-by-trial firing rates directly, rather than simulating individual spike trains, allowing precise control over key parameters such as field width, peak rate, baseline activity, and trial-level variability (Fig 5).

### Place Field Simulation

The activity of each neuron is modeled as a combination of a Gaussian tuning curve, a constant baseline, and additive noise on linear (1D) and two-dimensional (2D) spatial firing field (Fig 5A).

### Place field peak

A Gaussian curve was generated to simulate the place field of a neuron. This curve represents the neuron's firing rate as a function of spatial position along a linear track. The firing rate at position $x$ was defined by:

$$F(x) = A \cdot \exp\left(-\frac{(x - \mu)^2}{2\sigma^2}\right)$$

(1)

where $F(x)$ is the firing rate at position $x$, $A$ is the peak firing rate amplitude, $\mu$ is the center of the place field, and $\sigma$ is the standard deviation that determines the width of the field.

For the 2D case, the place field was modeled as a 2D Gaussian field. The firing rate at position $x,y$ was defined by:

$$F(x, y) = A \cdot \exp\left(-\frac{1}{2}\left[\left(\frac{x - \mu_x}{\sigma}\right)^2 + \left(\frac{y - \mu_y}{\sigma}\right)^2\right]\right)$$

(2)

where $(\mu_x, \mu_y)$ defines the center of the place field, and $\sigma$ determines the spatial spread equally across both dimensions. The spatial environment was discretized into equally spaced bins, with 50 bins in 1D and an $10 \times 10$ grid in 2D.

**Baseline.** A baseline firing rate was added to reflect background neural firing. This baseline, denoted $B(x)$ or $B(x,y)$ in 2D, was set to a constant value $B_0$ across all spatial bins to represent the neuron's firing rate in the absence of any spatial tuning or external stimuli:

$$B(x) = B_0$$

(3)

Note: $B(x,y) = B_0$ in 2D environments.

**Noise.** To simulate variability in neuronal firing, we added independent noise to each spatial bin. The noise at position $x$, denoted $N(x)$, was drawn from a Gaussian distribution:

$$N(x) \sim \mathcal{N}(0, \sigma_n^2)$$

(4)

where $N(x)$ is the noise at spatial bin $x$, and $\sigma n$ is the standard deviation controlling the magnitude of the noise. This noise was applied uniformly across all spatial bins, introducing random fluctuations that capture the inherent variability of neuronal activity. For the 2D case, independent noise was added to each spatial bin $(x, y)$:

$$N(x, y) \sim \mathcal{N}(0, \sigma_n^2)$$

(5)

where $N(x, y)$ denotes the noise at spatial location $(x, y)$, sampled independently across the two-dimensional grid.

**Simulated place field.** The final simulated firing rate was obtained by summing the Gaussian place field peak, the constant baseline firing rate, and the spatially distributed noise. The resulting firing rate at position $x$, denoted $G(x)$, was computed as:

$$G(x) = F(x) + N(x) + B(x)$$

(6)

For the 2D case, the firing rate at position $(x, y)$ was computed as:

$$G(x, y) = F(x, y) + N(x, y) + B(x, y)$$

(7)

where $N(x, y)$ represents spatially distributed noise across the two-dimensional environment. The combined signal, denoted $G(x)$ in one dimension and $G(x,y)$ in two dimensions, represents the neuron's firing rate as a function of spatial position, capturing both structured place-field activity and random noise fluctuations. The simulated place fields were subsequently used to compute spatial information scores and ANOVA statistics.

### Place field parameters

The tuning parameters - peak firing rate, place field width, baseline rate, and noise level - are independently manipulated between neurons (Fig 5B). Trial-to-trial variability is introduced through jittered field locations and a presence ratio that controls the probability of field expression (Fig 5C).

**Height.** To model variability in peak firing rates across neurons, we varied the amplitude parameter $A$ in the Gaussian tuning curve (Eq. 1), which sets the maximum firing rate at the place field center. We generated 1000 samples of $A$, each drawn from a uniform distribution over the range of [1,20] Hz.

**Width.** To simulate variability in spatial selectivity, we varied the place field width by adjusting the standard deviation parameter $\sigma$ in the Gaussian tuning curve (Eq. 1), defined over a 50-bin spatial track. Since $\sigma$ controls the spread of the tuning curve, it effectively determines half the width of the place field. Larger $\sigma$ values yield broader spatial tuning, while smaller values result in more sharply localized fields. $\sigma$ was sampled from 1,000 evenly spaced values between 1 and 20 spatial bins.

**Baseline.** To simulate background neuronal activity, we added a constant baseline firing rate $B_0$ across all spatial bins (Eq. 2). This represents non-selective, ongoing activity unrelated to spatial position. 1000 samples of $B_0$ were drawn uniformly between 0.5 and 5 Hz across the simulated population.

**Noise.** To introduce trial-to-trial variability, we added independent Gaussian noise to the firing rate at each spatial bin on every trial. The noise term $N(x)$ was drawn from a zero-mean Gaussian distribution with standard deviation $\sigma_n$ (Eq. 3). This noise was applied uniformly across space, simulating random fluctuations in neuronal firing. The noise level $\sigma_n$ was sampled uniformly between 0.5 and 5 Hz for 1000 neurons.

## Place field consistency

To model variability in place field position across trials, we introduced random shifts across simulated trials to the location of the Gaussian tuning curve. On each trial, 1000 samples of place field center $\mu$ was offset by a value drawn from a uniform distribution between 0 and 5 spatial bins. This introduces realistic trial-by-trial variability in the field location, simulating fluctuations commonly observed in empirical recordings.

## Presence ratio

To simulate variability in how consistently a neuron expressed its place field across trials, we used a presence ratio parameter. This ratio defined the proportion of trials in which the place field was active. On trials where the field was inactive, firing rates were set to baseline plus noise only. 1000 samples of presence ratios were sampled uniformly between 0.1 and 1 across simulated neurons.

## Statistical analysis

All firing rate computations and statistical analyses were performed using the `spiketools` Python package [67]. For both rodent and human datasets, firing rates were computed as spike counts per spatial bin normalized by occupancy time.

## Spatial information score

To quantify spatial tuning across the dataset, we computed the spatial information (SI) score for each neuron, which measures how much information a neuron's firing conveys about the subject's position. The SI score was defined as:

$$SI = \sum_x p(x) \cdot \frac{\gamma(x)}{\bar{\gamma}} \cdot \log_2 \left( \frac{\gamma(x)}{\bar{\gamma}} \right)$$

(8)

where $p(x)$ is the occupancy probability of spatial bin $x$, $\gamma(x)$ is the mean firing rate in bin $x$, and $\bar{\gamma}$ is the overall mean firing rate across all bins. SI was computed from each neuron's trial-averaged firing rate across spatial bins.

## ANOVA F-Statistics

To assess whether neurons were spatially modulated, we performed a one-way ANOVA on trial-by-bin firing rates for each neuron. For each trial, we computed the neuron's firing rate within each spatial bin, yielding a matrix of firing rates across

trials and positions. The one-way ANOVA tested whether the mean firing rate differed significantly across spatial bins, quantifying how much of the firing rate variance could be attributed to spatial location versus trial-to-trial variability. A high F-statistic indicated that a neuron's activity was reliably modulated by position across trials.

$$\text{firing rate} \sim C(\text{spatial bin}) \tag{9}$$

### Spatial information score threshold

To evaluate place cell classification, we applied two criteria. First, we used a commonly adopted threshold from rodent studies: neurons with SI greater than 0.25 bits/spike [24,28] were labeled as spatially selective.

### Permutation testing

To assess statistical significance for both spatial information (SI) scores and ANOVA F-statistics, we performed permutation testing using circular shuffling of spike trains within each session. For each neuron, spike times were circularly shifted along the session-wide spike train 1,000 times. This procedure preserved the overall temporal structure, firing rate, and autocorrelation of the spike train, while disrupting the relationship between spike timing and spatial position.

For each permutation, the shuffled spike train was used to recompute firing rates across spatial bins. A null distribution was then generated by recalculating the SI score and ANOVA F-statistic across the 1,000 shuffled iterations. A neuron was considered significantly spatially modulated if its observed score exceeded the 95th percentile of the corresponding null distribution (permutation-corrected, $p < 0.05$).

### Feature estimation

To characterize tuning properties and population structure across neurons, we extracted a set of quantitative features from each neuron's spatial firing profile. These features were designed to capture key aspects of spatial tuning strength, field structure, and trial-level consistency. For each neuron, we computed the following metrics:

- **Peak Firing Rate**: the maximum firing rate observed across all spatial bins.

- **Average Firing Rate**: the mean firing rate across all bins and trials.

- **Peak-to-average firing rate ratio**: a normalized measure of tuning sharpness, calculated as the ratio between peak and average firing rate.

- **Place Field Width**: the total number of spatial bins that exceeded a threshold firing rate (e.g., 20% of the peak), indicating contiguous spatial tuning.

- **Number of Place Fields**: the number of distinct spatially contiguous regions (fields) meeting minimum width and firing threshold criteria.

- **Place Field Consistency**: the proportion of trials in which the spatial bin of peak firing fell within a fixed window of ±3 bins around the neuron's overall peak bin, used to assess the reliability and consistency of spatial tuning across trials.

- **Presence ratio**: the proportion of trials in which the neuron exhibited non-zero firing in at least one bin within its identified place field region.

- **Even–Odd Correlation**: The Pearson correlation between the neuron's spatial firing rate maps computed separately from even- and odd-numbered trials, providing an estimate of trial-to-trial consistency in spatial tuning.

These features were concatenated into a feature vector for each neuron. The resulting matrix (neurons × features) served as input for dimensionality reduction using Principal Component Analysis (PCA).

## Dimensionality reduction with PCA

We performed Principal Component Analysis (PCA) on the neuron-by-feature matrix to identify low-dimensional structure in the tuning properties across the population. PCA finds a set of orthogonal axes (principal components) that explain the maximal variance in the data, enabling a compact representation of the dominant tuning patterns. The PCA projection allowed us to visualize and interpret the feature space, identify subpopulations with shared tuning characteristics, and explore the principal axes of variability in spatial encoding across neurons. Additionally, we examined the feature loadings on each component to estimate which place field properties most strongly contributed to each axis, and how these components related to standard spatial tuning metrics such as spatial information or ANOVA F-statistic.

## Materials descriptions and availability statements

### Project repository

This project is openly available through an online project repository, which includes all the code used for data pre-processing and analysis.

Project Repository: https://github.com/HSNPipeline/PlaceCellMethods

### Dataset

This project uses electrophysiological data collected from neurosurgical patients, as well as an open-access rat recording dataset from CRCNS.org: http://dx.doi.org/10.6080/K09G5JRZ.

The human single-neuron dataset was collected as part of a previously published study and is publicly available through OSF [14]: https://osf.io/dh3wv/metadata/osf.

To systematically evaluate place cell detection methods across species, we developed a custom simulation framework, SimPlaceCells, available at: https://github.com/HSNPipeline/SimPlaceCells.

### Software

All code used and developed for this project was written in the Python programming language. The code is openly available, licensed for reuse, and deposited in the project repository.

Management of the dataset was conducted using the Human Single Neuron (HSN) Pipeline:
https://github.com/HSNPipeline

Analyses of the single-neuron data were performed using the open-source Spiketools toolbox:
https://github.com/spiketools/spiketools

Literature searches and related resources were organized using LISC, an open-source Python module for literature analysis.
https://github.com/lisc-tools/lisc

## Supporting information

**S1 Fig. SI threshold and permutation methods identify different subsets of neurons in rats and humans.** Each cell shows the percentage of neurons classified as significant (*) or non-significant (N.S.) a) Rats: Comparison of SI classifications using a fixed threshold ($SI > 0.25$) and permutation-based significance testing. b) Same comparison in human neurons.
(TIFF)

**S2 Fig. Effect of smoothing on human place cell measures.** Each point represents a single neuron. a) ANOVA F-statistics computed from smoothed versus non-smoothed firing rate maps. b) Spatial information scores computed from smoothed versus non-smoothed data.
(TIFF)

**S3 Fig. Robustness of spatial tuning metrics across different spatial binning resolutions in rat and human recordings.** Rat data. a) ANOVA F-statistics comparisons: (Top) 20 vs. 10 spatial bins. (Bottom) 20 vs. 30 spatial bins. b) Spatial Information comparisons: (Top) 20 vs. 10 spatial bins. (Bottom) 20 vs. 30 spatial bins. Human data. c) ANOVA F-statistics comparisons: (Top) 40 vs. 20 spatial bins. (Bottom) 40 vs. 60 spatial bins d) Spatial Information (SI) comparisons: (Top) 40 vs. 20 spatial bins. (Bottom) 40 vs. 60 spatial bins. e–f) rat and g–h) human show the distributions of ANOVA F-statistics and Spatial Information across three spatial binning resolutions: 10, 20, 30 bins for rats and 20, 40, 60 bins for humans. i–j) rat and k–l) human summarize the average F-statistics and SI values for each binning resolution. m–n) show the percentage of significantly tuned neurons (y-axis) detected using ANOVA (blue) or SI (red) methods as a function of spatial binning (x-axis), for rats (m) and humans (n).
(TIFF)

**S4 Fig. Relationship between spatial tuning metrics and firing properties in rats, humans, and simulations.** Panels show how two tuning metrics: ANOVA F-statistics and SI relate to eight neural features: peak firing rate, average firing rate, peak-to-average ratio, place field width, number of place fields, place field consistency, presence ratio, and even–odd correlation. a-c, show data from rat recordings: a) distributions of each feature. b) F-statistics vs. features. c) SI vs. features. d-f) Human neurons. g-i) Simulated neurons. Each scatter plot includes the Pearson correlation coefficient r, quantifying the strength of association between tuning metrics and neural features.
(TIFF)

**S5 Fig. Pairwise correlations among neural features and their relationship to spatial tuning metrics in rats, humans, and simulations.** a) Rat dataset. b) Human dataset. c) Simulated dataset. Left, pairwise Pearson correlation coefficients between eight neural features: peak firing rate, average firing rate, peak-to-average ratio, place field width, number of place fields, place field consistency, presence ratio, and even–odd correlation. Middle, correlations between each feature and spatial information (SI). Right, correlations between each feature and ANOVA F-statistic. Color scale indicates the strength and direction of the correlation (red: positive; blue: negative).
(TIFF)

**S6 Fig. Principal component analysis of spatial coding metrics in human and simulated neurons.** a) Human neurons projected onto the first two principal components, colored by spatial information (SI) and ANOVA F-statistics. b) Same as a), for simulated data.
(TIFF)

**S7 Fig. PCA with feature visualization of neural features in rats, humans, and simulations.** Principal component analysis (PCA) was performed on eight firing-related features for a) rat. b) human. c) simulated datasets. Left, variance explained by each principal component; blue dots indicate cumulative explained variance, and the red dashed line marks the 90% variance threshold. Right, biplots showing PCA projections of neurons onto the first two principal components (PC1 and PC2), with feature loadings (red arrows) overlaid. The direction and length of the red vectors indicate each feature's contribution to the PCA axes.
(TIFF)

**S8 Fig. Impact of place field features on spatial information and ANOVA statistics in 2D Environments.** a) Simulated 2D firing rate maps illustrating variation across different width levels. b) Simulated 2D firing rate maps illustrating variation

across different noise levels. c) Spatial information (SI, red) and ANOVA F-statistics (blue) as a function of each place field parameter in 2D environments. d) Joint distributions of SI (y-axis) and ANOVA F-statistics (x-axis), with grayscale indicating values of the corresponding parameter in 2D environments.
(TIFF)

## Author contributions

**Conceptualization:** Weijia Zhang, Thomas Donoghue, Salman E Qasim, Joshua Jacobs.

**Data curation:** Weijia Zhang, Thomas Donoghue, Salman E Qasim.

**Formal analysis:** Weijia Zhang.

**Funding acquisition:** Joshua Jacobs.

**Investigation:** Thomas Donoghue, Joshua Jacobs.

**Methodology:** Weijia Zhang, Thomas Donoghue.

**Project administration:** Weijia Zhang, Thomas Donoghue.

**Supervision:** Thomas Donoghue, Salman E Qasim, Joshua Jacobs.

**Validation:** Thomas Donoghue.

**Visualization:** Weijia Zhang.

**Writing – original draft:** Weijia Zhang.

**Writing – review & editing:** Thomas Donoghue, Salman E Qasim, Joshua Jacobs.

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
