## [Decision Letter · Decision Letter 0]

2 Nov 2025

PCOMPBIOL-D-25-01752

Evaluating Place Cell Detection Methods in Rats and Humans – Implications for Cross-Species Spatial Coding

PLOS Computational Biology

Dear Dr. Zhang,

Thank you for submitting your manuscript to PLOS Computational Biology. After careful consideration, we feel that it has merit but does not fully meet PLOS Computational Biology's publication criteria as it currently stands. Therefore, we invite you to submit a revised version of the manuscript that addresses the points raised during the review process.

Please submit your revised manuscript within 60 days Jan 02 2026 11:59PM. If you will need more time than this to complete your revisions, please reply to this message or contact the journal office at ploscompbiol@plos.org. Please include the following items when submitting your revised manuscript:

We look forward to receiving your revised manuscript.

Kind regards,

Daniel Bush

Academic Editor

PLOS Computational Biology

Hugues Berry

Section Editor

PLOS Computational Biology

**Additional Editor Comments:**

In particular, the authors should ensure that their representation of the rodent place cell literature is accurate; consider extending their analyses to 2D firing rate maps; and include some consideration of more recent metrics based on information theoretic measures - as described in further detail in the reviewers comments below. Finally, given that rodent electrophysiology studies of spatial responses are much more mature, they should endeavour to clarify whether the ANOVA is ever a useful metric for identifying and describing place cell firing patterns.

**Journal Requirements:**

At this stage, the following Authors/Authors require contributions: Thomas Donoghue. Please ensure that the full contributions of each author are acknowledged in the "Add/Edit/Remove Authors" section of our submission form.

Potential Copyright Issues:

i) Figures 1, 2A, 3A, and 4. Please confirm whether you drew the images / clip-art within the figure panels by hand. If you did not draw the images, please provide (a) a link to the source of the images or icons and their license / terms of use; or (b) written permission from the copyright holder to publish the images or icons under our CC BY 4.0 license. Alternatively, you may replace the images with open source alternatives. See these open source resources you may use to replace images / clip-art:

2) If any authors received a salary from any of your funders, please state which authors and which funders..

7) Please send a completed 'Competing Interests' statement, including any COIs declared by your co-authors. If you have no competing interests to declare, please state "The authors have declared that no competing interests exist". Otherwise please declare all competing interests beginning with the statement "I have read the journal's policy and the authors of this manuscript have the following competing interests"

**Reviewers' comments:**

Reviewer's Responses to Questions

**Comments to the Authors:**

Reviewer #1: Summary

Zhang et al. compare two commonly used analytical approaches for identifying and characterizing spatially modulated neurons: spatial information content (SI) and analysis of variance (ANOVA), in both rodent and human hippocampal recordings. The authors also use a combination of real neural data and simulated firing patterns to examine how these methods differ in their sensitivity to firing field stability, spatial specificity, and overall detection rate. They further explore how analytical choices and thresholding criteria influence the classification of neurons as place cells, and how these methodological differences may affect cross-species comparisons. The topic addresses an important methodological gap in the field, as criteria for defining spatial cells are inconsistent across laboratories and species. The manuscript is well written, clearly illustrated, and makes a valuable contribution by encouraging a more principled approach to place cell classification. While there are no major analytical issues, and I agree with the authors' overall interpretation, I worry that the impact and general applicability of this work is overly specific.

Major comments

1. The introduction defines rat place cells as exhibiting “sharply localized and stable firing fields that tile the environment with high spatial precision.” It would strengthen the manuscript to provide a clearer operational definition of a place cell, as interpretations may differ across fields, especially between rodent electrophysiology and human cognitive neuroscience. For example, how should cells that fire broadly but consistently across the environment be classified? Likewise, should spatially precise but unstable cells be considered place cells? Establishing a consistent definition at the outset would help contextualize the comparisons between analytical methods.

2. The study focuses on recordings from linear track experiments, but most place cell studies are conducted in two-dimensional environments. It is not clear how well the ANOVA-based approach generalizes to two-dimensional data, or to unstructured exploration such as open-field foraging. Can the authors comment on whether ANOVA can be extended to such paradigms, and if not, whether this limits the method’s practical utility for broader applications?

3. The authors note that few studies have directly compared human and rodent hippocampal activity under comparable behavioral paradigms. However, in the present study, rodents actively explored a linear track, whereas humans were passively moved along a virtual track. Passive transport is known to disrupt spatial coding in rodents, which raises concerns about the comparability of the two datasets. This behavioral discrepancy may also explain the smaller proportion of human place cells observed and their reduced spatial specificity. The authors should address this limitation more explicitly and discuss its potential impact on the conclusions.

4. The field has begun to adopt more flexible analytical frameworks, such as generalized linear models (GLMs) that can account for collinearity and behavioral confounds, or information-theoretic measures like Kullback–Leibler divergence that are more robust to firing rate differences. Could the authors discuss how their findings might extend to or inform these more contemporary approaches? This would help situate the present work within the evolving methodological landscape.

5. The finding that SI is more sensitive to spatial specificity while ANOVA is more sensitive to trial-to-trial stability is interesting, but perhaps unsurprising given the mathematical properties of the two metrics. SI inherently reflects spatial contrast and signal-to-noise ratio, whereas ANOVA quantifies variance across trials. The authors could strengthen their argument by clarifying what conceptual or practical insight emerges from demonstrating this distinction.

Minor comments

1. Please include line numbers for ease of corrections.

2. The studies listed in Tables 1 and 2 should be cited in the reference list; otherwise, it is unclear which works the tables refer to.

3. The rationale for using a spatial information cutoff of 0.25 should be clarified. The authors later refer to this as a “commonly used fixed cutoff,” but most rodent studies in Table 1 use 0.5. Since nearly all cells exceed significance at this higher threshold (Fig. 2e), 0.25 seems unusually permissive and may bias the comparison against SI-based methods.

4. In the Figure 2 legend, the description for panel e incorrectly states “same as in e.”

5. The statement that “Spatial information is most sensitive to sharp, high-contrast firing fields and is strongly modulated by the ratio of peak to baseline activity” accurately describes a property that makes SI well suited to hippocampal place cells, though less so for neurons in other regions. However, the following point, that SI is “relatively insensitive to trial-to-trial instability and field dropout”, should be balanced by noting that stability is often explicitly verified in place cell studies by comparing maps before and after experimental manipulations.

6. In the section “Thresholding vs. Permutation Test and Effect of Analysis Parameters”, the authors may wish to reference Grieves (2023; PMID: 38150481), which also discusses how firing rate map construction parameters (e.g., bin size, kernel type) influence place cell classification.

7. The section “Place Cell Feature Estimates” appears to repeat conclusions already stated in the preceding paragraph; consider merging or shortening these sections to improve flow.

8. In the sentence “This organization is clarified by the feature loading vectors (Fig. 7c): the SI gradient aligns most strongly with peak-over-average firing rate (yellow) and runs opposite to place field width (cyan),” the colors do not match the figure (in Fig. 7, peak-over-average firing rate appears red and place field width magenta).

9. The authors might be interested in the approach used by Savelli et al. (2017; PMID: 28084992) where they not only using a shuffling procedure to determine the significance of a metric but also used bootstrapping to better calculate the actual metric. They used this for grid score, but it can be easily applied to spatial information content also. I think this approach could better take into account trial x trial instability, or be modified to patch that issue with SI.

Reviewer #2: This manuscript addresses an important methodological question in hippocampal research: how do different place cell detection methods affect cross-species comparisons? The authors systematically apply spatial information (SI) and ANOVA-based detection approaches to both rat and human datasets, supplemented by simulations with ground truth. They find that these metrics capture different aspects of spatial tuning - SI emphasises peak-to-average firing contrast while ANOVA captures trial-level consistency - and that methodological choices substantially influence which neurons are classified as place cells.

This systematic comparison of detection methods across species should be useful and important for the field, especially as human single-unit recording datasets become more prevalent.

Major Concern

1. Mischaracterisation of rodent literature undermines core message: Table 1 suggests only 3/24 rodent studies use stability measures as part of place cell classification, but this appears to significantly underrepresent standard practice. Recent rodent place cell research routinely uses some form of stability criteria to classify single-units as place cells. While the authors do not elaborate on how their rodent place cell studies were selected (“we subselected a sample of studies spanning a diversity of analytical pipelines”) - as a researcher in the rodent hippocampal field I do not deem their selection to be unbiased and representative. Indeed, conducting a very brief literature search myself I found the majority of papers I checked to use some stability criteria as part of their classification of place cells (e.g. Zhang et al 2024 Nature Comms, Blair et al 2023 eLIfe, Tanni et al., 2022 Current Biology to name but a few). This selective presentation undermines a key premise that rodent researchers ignore trial-level consistency picked up by ANOVA-based criteria. However, most crucially, this goes on to undermine one of the key messages, as defined in the Recommendations subsection: “Studies should consider the goals of their analyses and the likely properties of the place cells in their dataset to choose appropriate methods and/or consider applying multiple methods to capture different variants of putative spatial tuning.”. To summarise, I find the author’s presentation that rodent place cell research is typically conducted by classifying place cells according to a single criterion to be overly simplistic and misrepresentative. I would suggest a broad and unbiased literature search is used to accurately capture the current state-of-the-art

Minor Concerns

1. Confounding variables inadequately addressed: While the Discussion acknowledges differences between datasets (VR vs real-world, clinical constraints, anatomical sampling), these factors could explain much of the observed species differences. The interpretation that humans have "narrower" place cell distributions may reflect these experimental differences rather than biological ones.

2. Limited human dataset: With only 306 neurons from 19 patients, the human data may not capture the full diversity of hippocampal responses, particularly given the clinical constraints to participant selection and medication effects. This could be represented more clearly

3. Generalisability: Focus on linear track tasks limits conclusions about 2D spatial coding.

4. Self-citation rate: while not ignoring their contribution to the field, 12/48 references from the Jacobs lab represents quite a high self-citation rate (25%). Rather than removing references, which are used appropriately, I believe the paper would benefit from a more holistic representation of the hippocampal fields - both the human and rodent

Recommendation

This manuscript makes valuable contributions to standardising place cell detection across species. However, the characterisation of existing methods needs revision, particularly regarding stability measures in rodent studies. The authors should:

1. Conduct a more systematic literature review of rodent place cell methods

2. Revise claims about methodological differences between fields

3. More prominently discuss experimental confounds when interpreting species differences

4. Consider reducing self-citations where possible

With these revisions, this work would provide an important methodological resource for the field. The simulation framework and comparative analyses are valuable contributions that will help researchers make more informed choices about detection methods.

**Have the authors made all data and (if applicable) computational code underlying the findings in their manuscript fully available?**

Reviewer #1: Yes

Reviewer #2: Yes

PLOS authors have the option to publish the peer review history of their article (what does this mean?). If published, this will include your full peer review and any attached files.

Reviewer #1: No

Reviewer #2: No

**Figure resubmission:**
---

## [Decision Letter · Decision Letter 1]

19 Feb 2026

PCOMPBIOL-D-25-01752R1

Evaluating Place Cell Detection Methods in Rats and Humans – Implications for Cross-Species Spatial Coding

PLOS Computational Biology

Dear Dr. Zhang,

Thank you for submitting your manuscript to PLOS Computational Biology. After careful consideration, we feel that it has merit but does not fully meet PLOS Computational Biology's publication criteria as it currently stands. Therefore, we invite you to submit a revised version of the manuscript that addresses the points raised during the review process.

We look forward to receiving your revised manuscript.

Kind regards,

Daniel Bush

Section Editor

PLOS Computational Biology

Hugues Berry

Section Editor

PLOS Computational Biology

**Additional Editor Comments:**

The authors should address the few minor remaining comments from Reviewer #1 before we can proceed with publication. These final changes will be checked by the Editor - the manuscript will not be sent out for review again.

**Journal Requirements:**

1) We've observed that your S1-S2 files are duplicated, being both embedded in the main document and uploaded as SI Files. Please remove them from the main PDF and retain only the ones in the file inventory.

2) Please clarify the specific source name for the open-source clip-art in figures 1, 2A, 3A, 4

**Reviewers' comments:**

Reviewer's Responses to Questions

**Comments to the Authors:**

Reviewer #1: The authors have addressed my major comments. I have a couple of minor comments below, assuming the authors address these I would recommend publication in PLOS Computational Biology.

Minor comments

1. Fig S8 needs a dedicated methods section. This can build on the existing sections, but there needs to be a description of how the 2D maps were generated and what was used to calculate the SI/F-statistic values.

2. Fig S8 contains a number of typos, e.g. ‘F-Statistc’, ‘Place Field Cosnstency’. The colorbar labels are also easily confused as y-axis labels.

3. The Passive vs. Active Navigation section added to the discussion is informative, but the abstract will also need to be updated to reflect this limitation in interpretation. By adding, for example, “However we cannot rule out that the passive nature of the human navigation paradigm contributed to these effects.”

4. Previous minor comment 3: The added text is useful, but it needs to be clearer that the 0.25 b/s cutoff used is lower than most rodent studies and was lowered specifically so that a larger number of human cells pass criterion. The authors should also report how many human cells would meet the 0.5 b/s criterion.

Reviewer #2: The authors have made substantial revisions that address my major and minor concerns. The revised Table 1 now includes stability measures for the majority of the rodent studies surveyed, which provides a much more accurate representation of current practice in the field. The corresponding text revisions throughout the Abstract, Results, and Discussion appropriately reflect that rodent studies typically rely on multiple criteria for place-cell classification, rather than SI alone. I am satisfied that this addresses my primary concern.

The expanded Discussion sections on cross-species confounds (lines 508–580) are a welcome addition, particularly the explicit acknowledgement that factors such as anatomical coverage, VR-based navigation, medication effects, and clinical constraints may contribute to the narrower distribution of tuning scores observed in human recordings. The new subsections on anatomical differences, virtual environments, passive vs. active navigation, and clinical constraints provide a more balanced interpretation of the species comparison.

I note that the self-citation rate has been reduced from 25% to 18% through the addition of external references, which broadens the paper's coverage of the literature.

I am satisfied that the authors have adequately addressed my concerns and consider the manuscript suitable for publication.

**Have the authors made all data and (if applicable) computational code underlying the findings in their manuscript fully available?**

Reviewer #1: Yes

Reviewer #2: Yes

PLOS authors have the option to publish the peer review history of their article (what does this mean?). If published, this will include your full peer review and any attached files.

Reviewer #1: No

Reviewer #2: No

**Figure resubmission:**
---

## [Decision Letter · Decision Letter 2]

22 Apr 2026

PCOMPBIOL-D-25-01752R2

Evaluating Place Cell Detection Methods in Rats and Humans:  Implications for Cross-Species Spatial Coding

PLOS Computational Biology

Dear Dr. Zhang,

Thank you for submitting your manuscript to PLOS Computational Biology. After careful consideration, we feel that it has merit but does not fully meet PLOS Computational Biology's publication criteria as it currently stands. Therefore, we invite you to submit a revised version of the manuscript that addresses the points raised during the review process.

We look forward to receiving your revised manuscript.

Kind regards,

Daniel Bush

Section Editor

PLOS Computational Biology

Hugues Berry

Section Editor

PLOS Computational Biology

**Reviewers' comments:**

Reviewer's Responses to Questions

**Comments to the Authors:**

Reviewer #1: The authors have addressed many of my previous concerns; however, a fundamental issue remains regarding the abstract and the framing of the results. Specifically, the abstract fails to mention that the human and rodent datasets were collected under vastly different experimental paradigms (passive vs. active transport, and real-world vs. virtual reality).

As established in the literature (e.g. PMID: 25420065; PMID: 15390169), these variables are known to significantly modulate place cell dynamics. By omitting these differences from the abstract and the early discussion, the manuscript inadvertently implies that the observed differences are inherent species traits rather than products of the experimental design. The manuscript cannot make conclusions about cross-species differences, the methodological differences between species are too great. To prevent the data from being misconstrued or cited as a direct cross-species comparison of cell properties, it is essential that these methodological limitations are stated clearly in the abstract. I also insist that this should be acknowledged in the opening of the discussion to provide the necessary context for the results presented.

If these changes to the abstract and discussion are made I would recommend progression to publication.

**Have the authors made all data and (if applicable) computational code underlying the findings in their manuscript fully available?**

Reviewer #1: Yes

PLOS authors have the option to publish the peer review history of their article (what does this mean?). If published, this will include your full peer review and any attached files.

Reviewer #1: No

**Figure resubmission:**
---

## [Decision Letter · Decision Letter 3]

28 Apr 2026

Dear Miss Zhang,

We are pleased to inform you that your manuscript 'Evaluating Place Cell Detection Methods in Rats and Humans:  Implications for Cross-Species Spatial Coding' has been provisionally accepted for publication in PLOS Computational Biology.

Best regards,

Daniel Bush

Section Editor

PLOS Computational Biology

Hugues Berry

Section Editor

PLOS Computational Biology

Reviewer's Responses to Questions

**Comments to the Authors:**

Reviewer #1: I would like to thank the authors for their responsiveness and for incorporating the suggested changes into the abstract and discussion. These additions provide essential context that significantly strengthens the impact and accuracy of the work. This manuscript provides a very useful roadmap for researchers navigating the complexities of place cell detection across different species and analysis pipelines and I look forward to seeing this work published and cited. Congratulations on a rigorous and interesting study!

**Have the authors made all data and (if applicable) computational code underlying the findings in their manuscript fully available?**

Reviewer #1: Yes

PLOS authors have the option to publish the peer review history of their article (what does this mean?). If published, this will include your full peer review and any attached files.

Reviewer #1: No

---

## [Editor Report · Acceptance letter]

PCOMPBIOL-D-25-01752R3

Evaluating Place Cell Detection Methods in Rats and Humans:  Implications for Cross-Species Spatial Coding

Dear Dr Zhang,

I am pleased to inform you that your manuscript has been formally accepted for publication in PLOS Computational Biology. Your manuscript is now with our production department and you will be notified of the publication date in due course.

With kind regards,

Anita Estes
